# Towards Fair In-Context Learning with Tabular Foundation Models

**Patrik Kenfack**
**ÉTS Montréal**
**Mila - Quebec AI Institute**
`patrik-joslin.kenfack.1@ens.etsmtl.ca`

**Samira Ebrahimi Kahou**
**University of Calgary**
**Mila - Quebec AI Institute, CIFAR**
`samira.ebrahimikahou@ucalgary.ca`

**Ulrich Aïvodji**
**ÉTS Montréal**
**Mila - Quebec AI Institute**
`ulrich.aivodji@etsmtl.ca`

**Reviewed on OpenReview:** `https://openreview.net/forum?id=AsBhwDOsqo`

## Abstract

Transformer-based tabular foundation models have recently demonstrated promising in-context learning (ICL) performance on structured data, emerging as competitive alternatives to gradient-boosted trees. However, the fairness implications of this new paradigm remain largely unexplored. We present the first investigation of fairness in tabular ICL, evaluating three recently proposed foundation models—TabPFNv2, TabICL, and TabDPT—on multiple benchmark datasets. To mitigate biases, we explore three pre-processing fairness-enhancing methods: correlation removal (decorrelating input features from the sensitive attribute), group-balanced sample selection (ensuring equal representation of protected groups in context examples), and uncertainty-based sample selection (prioritizing context examples with high sensitive-attribute prediction uncertainty). Our experiments show that the uncertainty-based strategy consistently improves group fairness metrics (e.g., demographic parity, equalized odds, and equal opportunity) with minimal impact on predictive accuracy. We release our code to facilitate reproducibility (https://github.com/patrikken/Fair-TabICL).

## 1 Introduction

Tabular data, represented in rows and columns, is a data modality widely used for prediction tasks in domains such as finance and healthcare (Asuncion et al., 2007). Tree-based models such as XGboost (Chen et al., 2015) and Gradient-Boosted Trees (Ke et al., 2017) have shown the strongest generalization performance on tabular data. Recently, with the emergence of foundation models, Deep Learning (DL) based models have challenged the dominance of tree-based models (Hollmann et al., 2025). Foundation models are models pretrained on vast datasets, without a specific task in mind, and they can be adapted across a wide range of downstream tasks. Large language models (LLMs) such as GPT-3 are common examples of foundation models, and they have demonstrated emerging capabilities such as in-context learning (ICL) with few labelled data (Brown et al., 2020). In-context learning (ICL) has primarily been applied to natural language tasks using large

language models (LLMs). For example, in text classification, labeled examples are formatted as textual demonstrations and provided as context to a language model, enabling it to predict the label of a new instance without any parameter updates or fine-tuning (Radford et al., 2019; Brown et al., 2020). More recently, efforts have extended ICL to tabular data by serializing table rows into text or sentences (Hegselmann et al., 2023). However, since LLMs are not pretrained to model the complex structural relationships inherent in tabular data—such as interactions between rows and columns—their performance on large-scale tabular tasks still lags behind tree-based methods (Hegselmann et al., 2023).

Alternatively, recent work has proposed foundation models explicitly tailored for tabular data, achieving competitive performance with tree-based models while reducing the need for extensive model selection and hyperparameter tuning. For instance, TabPFN (Hollmann et al., 2022) is a transformer-based model pretrained on synthetic datasets, and its successor, TabPFNv2 (Hollmann et al., 2025), extends support to larger datasets with up to 10k samples. Similarly, TabICL (Qu et al., 2025), also pretrained on synthetic data, scales to datasets with up to 500k samples. These models leverage synthetic pretraining to encode a wide range of statistical priors, enabling effective target inference from in-context examples.

To better reflect the priors found in real-world datasets, other models incorporate real data during pretraining. Tabular Discriminative Pre-trained Transformer (TabDPT) (Ma et al., 2024) is pretrained directly on real-world datasets, while Real-TabPFN (Garg et al., 2025) builds on synthetic pretraining with additional fine-tuning on real data. These approaches generally yield improved performance by aligning the learned representations more closely with real-world data distributions.

Given the strong performance and in-context learning capabilities of tabular foundation models, we will likely see widespread adoption in real-world decision-making tasks. This shift could mark a turning point in how tabular data problems are approached. However, the use of ICL-based models in high-stakes domains—such as healthcare, finance, or criminal justice—raises important ethical concerns. In particular, it is critical to assess their potential to perpetuate or even amplify existing social biases. Traditional machine learning models have already been shown to replicate biases present in the data (Mehrabi et al., 2022), and recent studies indicate that LLM-based ICL can also produce biased predictions (Hu et al., 2024; Bhaila et al., 2024). However, these studies rely on serialized representations of tabular data and therefore inherit the limitations of LLMs in handling tabular structures (Ma et al., 2024).

This paper investigates the fairness of ICL prediction using transformer-based tabular foundation models. First, our study reveals, perhaps unsurprisingly, that while these models focus on improving prediction accuracy, they can also amplify bias. Motivated by recent studies on the sensitivity of ICL performance—in terms of fairness and accuracy— to demonstration selection, we aim to address the following research question: *What in-context selection/transformation method can improve the fairness of ICL predictions?*

In the fairness literature, several metrics have been proposed to measure fairness at the group or individual levels (Dwork et al., 2012). In this work, we focus on widely used group fairness notions, including demographic parity, equalized odds, and equal opportunity (Hardt et al., 2016). This group metrics measure the performance disparity across different demographic groups while we acknowledge that other fairness metrics, beyond group fairness, such as individual or counterfactual fairness, could be used depending on the use case. To achieve these fairness notions, several fairness-enhancing methods have been proposed. They are generally grouped into three categories: pre-processing, in-processing, and post-processing (Mehrabi et al., 2022) methods. Projecting these categories into the ICL paradigm, pre-processing methods perform demonstration transformation or selection before predicting in context (Hu et al., 2024). In-processing methods would fine-tune or retrain the foundation model with fairness constraints (Robertson et al., 2024). Post-processing methods would alter the ICL predictions to improve a given fairness metric (Hardt et al., 2016). Pre- and Post-processing methods are more computationally friendly since they do not require model updates. This motivates our choice to focus on the pre-processing techniques and leave post-processing interventions for future exploration. More specifically, we propose and investigate three pre-processing fairness interventions: (i) Correlation Remover (Feldman et al., 2015), a method that alters each input feature to reduce their correlation with the sensitive attribute; (ii) group-balanced[1] in-context selection, ensures that the in-context set is group-balanced; (iii) Uncertainty-based in-context selection, estimates the uncertainty of predicting the

---

[1] Underlined represents the method's name throughout the paper and in the results.

sensitive attribute of in-context samples and only selects samples with uncertain predictions. We performed intensive experiments on eight fairness benchmark datasets to investigate the effectiveness of each method in terms of fairness and accuracy. Our results reveal that the uncertainty-based method can provide better fairness performance across datasets, fairness metrics, and foundational models, with marginal impact on accuracy. Our contribution can be summarized as follows:

- While most existing studies focus on fair ICL with serialized tabular data, we provide, to our knowledge, the first investigation into preprocessing methods for fair prediction in ICL using transformer-based tabular foundation models.

- We propose and investigate three pre-processing intervention methods to enforce fair ICL predictions. These methods aim to reduce the information about the sensitive attributes of in-context samples. We demonstrate that uncertainty-based in-context sample selection can significantly improve the fairness of ICL predictions with a slight drop in utility, e.g., accuracy.

- We perform extensive experiments on a broad range of start-of-the-art fairness benchmarks and provide insights into contexts where a given fairness intervention performs best in terms of fairness-utility tradeoff.

- We release the code to ease reproduction of the results and help researchers and practitioners integrate the proposed methods.

## 2 Related works

**Fairness.** Numerous methods have been developed to enforce group fairness in classical machine learning models (Mehrabi et al., 2022; Mbiazi et al., 2023; Kenfack et al., 2024a). These methods are often categorized as pre-processing, in-processing, or post-processing approaches. Model-agnostic methods in the pre-processing category typically modify or reweight the input data to reduce information correlated with sensitive attributes (Madras et al., 2018; Creager et al., 2019; Kamiran & Calders, 2012; Celis et al., 2020; Balunović et al., 2021; Feldman et al., 2015). In contrast, post-processing techniques adjust the model's prediction outcomes after training to satisfy fairness constraints (Hardt et al., 2016; Petersen et al., 2021). Finally, in-processing approaches embed fairness constraints directly into the training objective (Agarwal et al., 2018; Zhang et al., 2018; Roh et al., 2020). Unlike prior work, our approach focuses on pre-processing interventions applied in in-context learning (ICL) settings, where downstream predictions are made by foundation models without any model updates. We emphasize that model-agnostic methods—particularly pre- and post-processing—are especially suitable in ICL because they do not rely on access to or retraining of the model. However, their effectiveness in this setting, especially for tabular foundation models, remains largely unexplored and is the focus of our evaluation.

**Tabular Foundation Models.** In-context learning with tabular foundation models presents a notable advantage over traditional machine learning approaches by enabling models to adapt dynamically to new data without the need for retraining (Hollmann et al., 2022; Qu et al., 2025; Hollmann et al., 2025). Conventional ML methods typically depend on predefined training datasets, meaning that any alteration in the data or task necessitates a time-consuming and resource-intensive retraining process. In contrast, tabular foundation models utilize in-context learning to execute tasks based on the specific context of the data provided at inference time. This allows these models to interpret and process new tabular data with minimal prior preparation, facilitating more flexible and efficient decision-making (Hollmann et al., 2022). The advantages of this approach are particularly apparent in scenarios where data distributions change over time or when models must quickly adjust to various data tasks without undergoing retraining. Thus, as in-context learning emerges as a powerful tool for real-time, adaptive predictions in complex and dynamic environments, assessing and mitigating biases in the prediction can make its use more socially acceptable. Existing models are pre-trained using synthetic (Qu et al., 2025; Hollmann et al., 2022) or real-world data (Ma et al., 2024). Pre-training on real-world data often provides competitive or better performance, and the performance of synthetic pre-trained tabular foundation models can be boosted with continued pre-training on real-world data (Garg et al., 2025). In this work, we consider tabular foundation models pretrained on synthetic (Hollmann et al.,

2025; Qu et al., 2025) and real-world data (Ma et al., 2024) and assess the fairness implications of each pretraining strategy.

**Fairness in ICL.** Fairness in in-context learning has primarily been studied in the context of large language models (LLMs) applied to tabular data serialized as text(Bhaila et al., 2024; Hu et al., 2024; Ma et al., 2023). For instance, Hu et al. (2024) explore group-aware sampling strategies, finding that prioritizing minority group demonstrations improves fairness outcomes. Similarly, Bhaila et al. (2024) propose a data augmentation technique that guides demonstration selection to reduce bias during inference. While related in spirit, our approach differs in two key ways: (1) we focus on numerical tabular foundation models (rather than LLMs), and (2) our uncertainty-based selection method aims to reduce model reliance on sensitive attributes rather than optimize informativeness per se. Prior uncertainty-driven methods(Mavromatis et al., 2023; Kung et al., 2023) focus on selecting informative examples under a labeling budget, whereas we use uncertainty to guide fair demonstration selection. It is also important to highlight that LLMs, though flexible, are not optimized for tabular data and often perform worse than specialized numeric models, such as gradient-boosted trees (Hegselmann et al., 2023). As such, our work fills a key gap by investigating fairness interventions tailored to numeric tabular foundation models. To our knowledge, this is the first study to evaluate pre-processing fairness methods in this emerging model class.

**Fairness-Aware Tabular Foundation Models.** Recent work by Robertson et al. (2024) introduced FairPFN, a TabPFN-like model trained to suppress the causal effect of sensitive attributes during pretraining. Their approach seeks counterfactual fairness (Kusner et al., 2017), ensuring that model predictions remain invariant when sensitive attributes are counterfactually changed. Our work is conceptually distinct in two respects. First, we do not require model retraining and rely on model-agnostic pre-processing methods, making our approach broadly applicable to any pretrained tabular foundation model. Second, we aim for group fairness, focusing on improving performance disparities between subgroups, rather than enforcing counterfactual invariance at the individual level.

## 3 Methodology

**Problem Setup** We consider a classification task with the given training data $\mathcal{D} = \{(x_i, y_i, s_i)\}_{i=1}^{N}$ where $x_i$ is an input feature vector, $y_i$ is the corresponding class label, and $s_i$ the corresponding demographic group. The goal is to obtain a classifier $f$, via ICL, to accurately predict the target $y$ given a sample $x$ while being *fair* w.r.t. demographic information $s$. Several metrics have been proposed to measure fairness at the group or individual levels (Dwork et al., 2012). In this work, we focus on group fairness notions, measuring the performance disparity across different demographic groups, i.e., demographic parity, equalized odds, and equal opportunity (Hardt et al., 2016). A detailed description of these fairness metrics can be found in Appendix B.1.

This section presents three pre-processing techniques proposed in this work to ensure fairer ICL inference on tabular data. In particular, we consider *correlation remover* (Feldman et al., 2015), group-balanced demonstration selection, and uncertainty-based demonstration selection.

### 3.1 In-context Samples Transformation

Correlation remover (Feldman et al., 2015; Bird et al., 2020) is a preprocessing method that reduces the correlation between the sensitive and non-sensitive attributes before fitting the model. More specifically, a linear transformation is applied to each non-sensitive feature to reduce its correlation with the sensitive feature. We use the correlation remover as a preprocessing step over the training (in-context example) and testing sets before performing in-context prediction. Ultimately, transforming input features to reduce their linear dependency on the sensitive feature can reduce the reliance on sensitive features in the downstream models. However, as we will see in Section 4.5, nonlinear and complex downstream models can still infer the nonlinear dependencies over the sensitive feature and provide unfair results. A more detailed description of correlation remover can be found in Appendix B.2.

### 3.2  In-context Samples Selection

In this work, we posit that in-context sample selection can have a significant impact on the fairness of ICL prediction. We analyze two demonstration selection methods that can improve the fairness of ICL predictions without model update.

#### 3.2.1  Group-balanced demonstration set selection.

Representation bias is a common source of bias in machine learning models (Mehrabi et al., 2022). It occurs when the collected training data does not reflect the demographic diversity of the population. As a result, some demographic subgroups are under-represented, if not represented at all. Recent studies have demonstrated the benefits of group-balanced training data on the fairness properties of the downstream model. Several methods have been proposed to mitigate representation bias in the data, including *subsampling the majority group* or *reweighting the training data* based on group proportions (Kamiran & Calders, 2012; Celis et al., 2020). In this paper, we focus on *subsampling* since current tabular foundation models do not handle sample weights (Hollmann et al., 2025; Qu et al., 2025). We perform ICL with a group-balanced demonstration set sampling from each group uniformly at random. When the demonstration set size does allow equal group representation, we subsample the majority group at random. A similar strategy is employed by (Hu et al., 2024) to select demonstrations for few-shot ICL prediction with LLMs. In this paper, we evaluate the effectiveness of this fairness intervention with tabular foundation models instead of using LLMs on serialized tabular data.

#### 3.2.2  Uncertainty-based demonstration set selection.

Kenfack et al. (2024b) demonstrated that models trained without fairness constraints can have better fairness properties when the training data consists of samples with uncertain sensitive attributes. We hypothesize that *the uncertainty of the sensitive attribute prediction can be a good measure to select demonstrations that improve the fairness of in-context predictions.* To validate this, we measure the uncertainty of predicting the sensitive attribute in the demonstration set and use samples with high uncertainty for in-context learning. We focus on conformal prediction (Shafer & Vovk, 2008; Vovk et al., 2005) as uncertainty measure since it provides strong theoretical guarantees for the coverage. Instead of returning a single label, a conformal predictor returns a prediction set containing the true label with a probability of at least $1 - \epsilon$, with $\epsilon$ being a user-defined coverage parameter of the conformal prediction (Angelopoulos et al., 2023). For example, setting $\epsilon = 0.1$ ensures the prediction set contains the true sensitive attribute value with at least 90% probability. Specifically, samples with prediction sets containing more than one value are uncertain. Intuitively, the coverage parameter $\epsilon$ controls the fairness-utility tradeoff, with $\epsilon \approx 1$ meaning no fairness intervention where all the datapoints are used and $\epsilon \approx 0$ meaning maximal fairness intervention where only uncertain samples are included in the in-context examples. We show in Section 4.2.1 how the coverage parameter $\epsilon$ of the conformal predictor consistently controls the tradeoff between accuracy and fairness across fairness metrics and downstream foundational models.

Since conformal prediction is model agnostic, we considered both classical methods, e.g., Logistic Regression (LR), and foundation models, e.g., TabPFN for training the sensitive attribute classifier to measure the prediction uncertainties. Note that the method could be applied using other uncertainty measures, such as Monte Carlo dropout and confidence interval (Kenfack et al., 2024b). We focus on conformal prediction due to its rigorous theoretical guarantees, and it does not require a hyperparameter to threshold the level from which a prediction is considered uncertain (Angelopoulos et al., 2023). More details about uncertainty measurement with conformal prediction can be found in Appendix B.3.

### 3.3  In-Context Prediction

After performing demonstration selection or transformation using the fairness intervention methods presented previously, we pass them through the tabular foundation model as in-context examples for predicting class labels on the test set. In this paper, we consider three tabular foundation models: TabDPT (Ma et al., 2024), TabICL (Qu et al., 2025), and TabPFNv2 (Hollmann et al., 2025). While all these models use Transformer

architectures as their backbones and are pretrained on a large amount of datasets, TabICL and TabPFN use only synthetic data generated from structured causal networks, and TabDPT uses real-world data. The current version of TabPFN can handle a maximum of 10k samples with 500 features, and TabICL can handle up to 500K samples. We randomly subsample in-context examples when the context set size exceeds the maximum number of samples allowed by the foundation model being evaluated.

# 4 Experiments

We describe the experimental setup and perform intensive experiments to answer the following research questions:

- (R1) Does group-balanced demonstration selection and correlation remover effectively reduce information about the sensitive attribute and improve the fairness of ICL?

- (R2) What fairness intervention provides a better fairness-utility tradeoff?

- (R3) What foundation tabular model can provide a better fairness-utility tradeoff?

## 4.1 Experimental Setup

**Datasets**  We experiment on tasks from `folktables` (Ding et al., 2022), which contains data extracted from the American Community Survey (ACS) Public Use Microdata Sample (PUMS). More specifically, we experiment with the following ACS PUMS tasks: ACSIncome, ACSMobility, ACSTravelTime, ACSEmployment, ACSPublicCoverage. These tasks reflect a range of real-world predictive challenges with fairness concerns. We use the data of the year 2018 from the state of Alabama (AL), which is one of the states with the largest fairness violation (Ding et al., 2022). A limitation of the ACS PUMS datasets is that they are US-centric; we diversify the experimental setup by including other tasks and datasets. Specifically, we also experiment on the other tabular datasets and tasks including: Diabetes (Gardner et al., 2023), German Credit (Frank, 2010), and CelebA (Liu et al., 2018). More details about each dataset, including the sensitive attributes used, the number of samples, and the number of features, can be found in the Appendix A.

**Metrics**  We measure the utility of the model using accuracy, and measure fairness using three group fairness metrics, i.e., Demographic Parity ($\Delta$DP), Equal Opportunity ($\Delta$EOP), and Equalized Odds ($\Delta$EOD). More details about group fairness metrics can be found in B.1.

**Baselines**  We evaluate ICL under four in-context selection strategies, comparing both fairness and accuracy:

- `Vanilla`: randomly selects in-context examples from the training set without fairness considerations.

- **Group Balanced**: samples in-context examples to maintain equal group ratios, uniformly downsampling the majority group across runs.

- `Correlation Remover` (CR): applies correlation-removal (Feldman et al., 2015) to reduce dependence between sensitive and non-sensitive features in both in-context and test examples. We use the `fairlearn` implementation (Bird et al., 2020) with varying values of $\alpha$, where $\alpha = 1$ enforces maximal fairness.

- `Uncertain`: selects in-context examples with high uncertainty in sensitive attribute prediction. We estimate uncertainty using conformal prediction (Cordier et al., 2023) as implemented in `Mapie` (Taquet et al., 2022), varying the coverage parameter $\epsilon$ to control the fairness–utility tradeoff. We study two variants of sensitive attribute classifier: one with a Logistic Regression classifier (`Uncertain+LR`) and one with a foundation model (`Uncertain+TabPFN`).

**Evaluation** For evaluation, we held out 20% of the data to train the sensitive attribute classifier used in the uncertainty-based baseline. The remaining data were split into 80% for training and 20% for testing. We applied the fairness interventions to the training set to derive in-context samples, and reported fairness and accuracy on the test set, averaged over three random seeds. This setup provides robustness, as each data point can appear as a test sample or an in-context sample across seeds. As noted above, we use TabICL, TabDPT, and TabPFN with their default parameters.

## 4.2 Results and Discussion

We evaluate fairness in ICL predictions with tabular foundation models along several dimensions. First, we compare the baseline methods in terms of fairness and accuracy. For interventions with a tunable fairness–utility tradeoff, we vary the corresponding hyperparameter and compare their Pareto fronts. While we use accuracy as the main utility metric, we verify in the Appendix that results are consistent when using ROC AUC. Next, we compare foundation models under the best-performing intervention. Finally, we conduct an ablation on the effect of in-context set size and analyze the failure case of the correlation-removal method.

### 4.2.1 Fairness-Utility Tradeoff

Figure 1 shows the fairness–utility Pareto fronts on ACSIncome, ACSMobility, and ACSTravelTime datasets using TabPFN as the foundation model. We compare the `Vanilla` baseline, `Group Balanced`, `Correlation Remover`, and the two variants of our uncertainty-based method (`Uncertain+LR` and `Uncertain+TabPFN`). For `Correlation Remover` and `Uncertain`, we vary the tradeoff parameters $\alpha$ and $\epsilon$ over $[0, 1]$, with each shaded point corresponding to a parameter setting. The main findings, consistent across datasets and models, are as follows:

- (R1) The `Group Balanced` method yields only marginal fairness gains over `Vanilla` ICL, with similar utility. This suggests that representation bias alone does not explain observed disparities; even with group-balanced contexts, subgroup performance gaps remain, likely due to other sources of bias such as historical, algorithmic, or measurement bias (Mehrabi et al., 2022).

- (R1, CR failure) Surprisingly, `Correlation Remover` often exacerbates unfairness. For example, on ACSIncome, demographic parity worsens from 0.14% to 0.26% at $\alpha = 1$ (see Figure 1). We attribute this to bias amplification due to sensitive attribute leakage, as discussed in Section 4.5.

- (R2) In contrast, both variants of `Uncertain` consistently achieve Pareto-dominant points. They offer stable control over the fairness–accuracy tradeoff across $\epsilon$ values, enabling significant fairness improvements with modest accuracy loss. The reduced accuracy stems mainly from smaller in-context sets, since excluding low-uncertainty samples decreases the size of the in-context set, which can reduce accuracy as we discuss in Section 4.3).

- (R2, Uncertainty variants) Among the two uncertainty-based methods with different model classes, `Uncertain+TabPFN` typically yields stronger Pareto fronts, indicating that TabPFN provides more reliable conformal predictions than logistic regression, leading to better fairness-aware sample selection.

We observe similar patterns on other datasets. Additional results are reported in the Appendix: Figure 6 for other datasets with TabPFN; Figures 7 and 9 for TabICL; and Figures 8 and 10 for TabDPT.

### 4.2.2 Accuracy of sensitive attribute reconstruction

We assess whether foundation models can reconstruct the sensitive attribute after different fairness interventions. Specifically, we perform ICL to predict the sensitive attribute and measure test accuracy. Effective debiasing should lower this accuracy, ideally approaching random guessing (50%).

Table 1 shows that the Group Balanced intervention only marginally reduces reconstruction accuracy for TabPFN and TabICL on ACS datasets. This limited effect is expected, since ACS is nearly group-balanced for gender (Ding et al., 2022), e.g., for the state of Alabama, the gender distribution is 51.96% females,

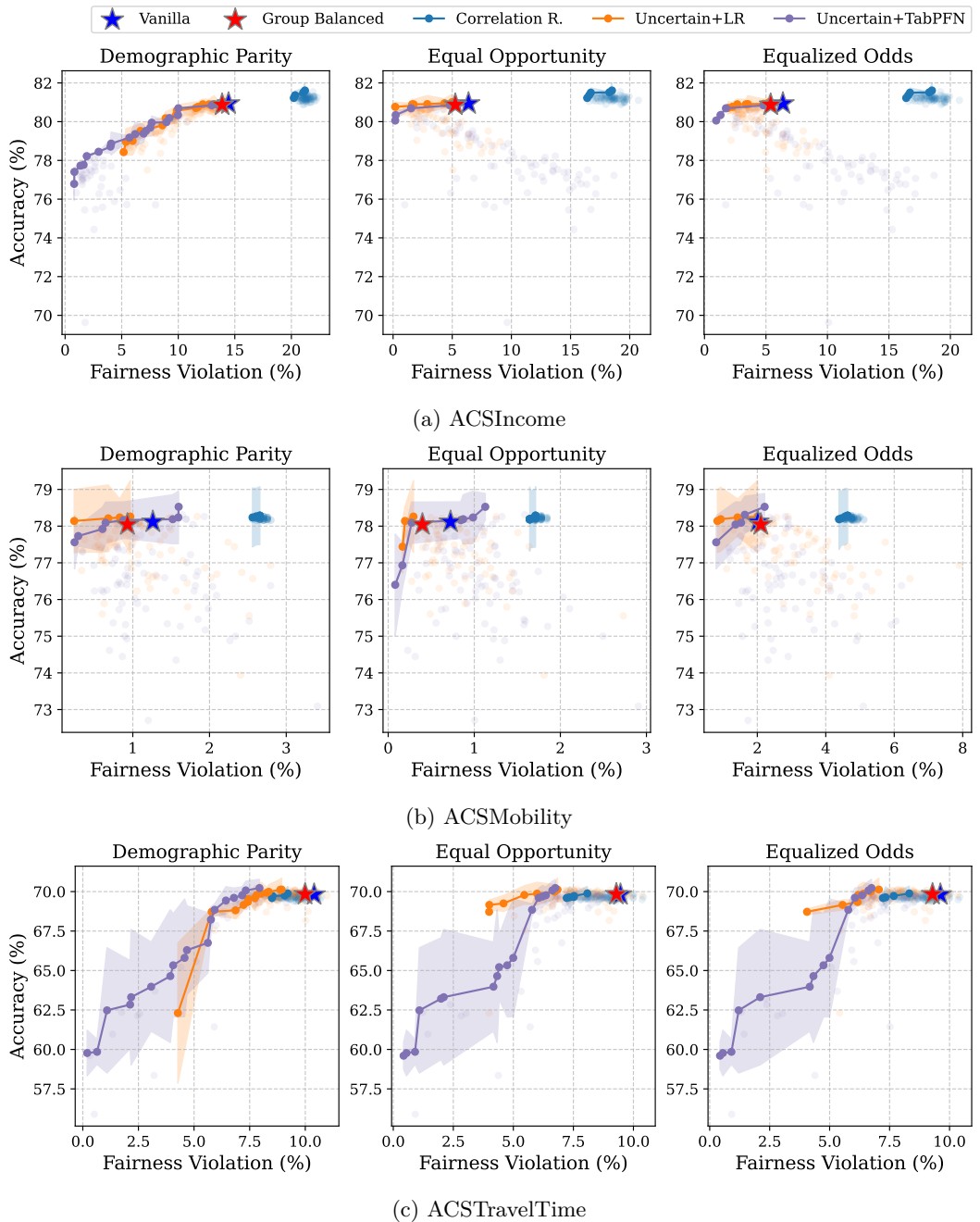

(a) ACSIncome

(b) ACSMobility

(c) ACSTravelTime

Figure 1: Comparing the fairness-utility Pareto-front of different fairness interventions using TabPFN on the ACSIncome, ACSMobility, and ACSTravelTime datasets.

48.04% males. In contrast, the Diabetes dataset is highly imbalanced by race, and group balancing reduces reconstruction accuracy from 80.2% for the vanilla baseline to 66.2%.

For `Correlation Remover`, we observe the opposite effect: reconstruction accuracy rises to nearly 100%. We identify this as the source of bias amplification. Transformer-based foundation models, pretrained on synthetic data with diverse structural priors, appear to detect and exploit the transformations applied by `Correlation Remover`. Because each non-sensitive feature is modified using the sensitive attribute, sensitive information leaks into the transformed features. The models leverage this leakage, relying on hidden sensitive

| Dataset | ICL Method | TabPFN | | TabICL | |
|---|---|---|---|---|---|
| | | Accuracy ↓ | F1 Score ↓ | Accuracy ↓ | F1 Score ↓ |
| ACSIncome | Vanilla | $77.2_{\pm0.5}$ | $78.4_{\pm0.3}$ | $75.0_{\pm0.5}$ | $76.2_{\pm0.3}$ |
| | Group Balanced | $77.1_{\pm0.5}$ | $77.8_{\pm0.2}$ | $75.0_{\pm0.4}$ | $75.5_{\pm0.1}$ |
| | Correlation R. | $100.0_{\pm0.0}$ | $100.0_{\pm0.0}$ | $99.9_{\pm0.0}$ | $99.9_{\pm0.0}$ |
| | Uncertain+LR | $74.7_{\pm1.3}$ | $76.7_{\pm0.6}$ | $74.9_{\pm0.5}$ | $76.0_{\pm0.4}$ |
| | Uncertain+TabPFN | $\mathbf{51.3}_{\pm2.3}$ | $\mathbf{66.1}_{\pm4.4}$ | $\mathbf{71.7}_{\pm0.4}$ | $\mathbf{73.6}_{\pm0.6}$ |
| ACSTravelTime | Vanilla | $75.9_{\pm0.4}$ | $77.5_{\pm0.5}$ | $72.8_{\pm0.4}$ | $73.8_{\pm0.5}$ |
| | Group Balanced | $75.9_{\pm0.5}$ | $77.0_{\pm0.5}$ | $72.5_{\pm0.6}$ | $72.6_{\pm0.6}$ |
| | Correlation R. | $100.0_{\pm0.0}$ | $100.0_{\pm0.0}$ | $100.0_{\pm0.0}$ | $100.0_{\pm0.0}$ |
| | Uncertain+LR | $75.9_{\pm0.6}$ | $77.8_{\pm0.5}$ | $72.8_{\pm0.5}$ | $74.3_{\pm0.5}$ |
| | Uncertain+TabPFN | $\mathbf{74.6}_{\pm1.1}$ | $\mathbf{75.7}_{\pm1.7}$ | $\mathbf{67.4}_{\pm1.6}$ | $\mathbf{66.2}_{\pm2.4}$ |
| ACSPublicCoverage | Vanilla | $91.4_{\pm0.2}$ | $88.9_{\pm0.2}$ | $91.5_{\pm0.1}$ | $89.2_{\pm0.2}$ |
| | Group Balanced | $91.1_{\pm0.4}$ | $89.0_{\pm0.3}$ | $90.9_{\pm0.3}$ | $88.9_{\pm0.4}$ |
| | Correlation R. | $100.0_{\pm0.0}$ | $100.0_{\pm0.0}$ | $100.0_{\pm0.0}$ | $100.0_{\pm0.0}$ |
| | Uncertain+LR | $56.4_{\pm10.0}$ | $\mathbf{55.3}_{\pm4.0}$ | $57.7_{\pm11.9}$ | $60.0_{\pm3.6}$ |
| | Uncertain+TabPFN | $\mathbf{42.5}_{\pm0.9}$ | $58.7_{\pm0.6}$ | $\mathbf{42.7}_{\pm1.1}$ | $\mathbf{58.6}_{\pm0.6}$ |
| ACSEmployment | Vanilla | $64.0_{\pm0.4}$ | $62.0_{\pm1.8}$ | $65.0_{\pm0.3}$ | $62.2_{\pm1.4}$ |
| | Group Balanced | $64.0_{\pm0.5}$ | $65.0_{\pm1.0}$ | $64.8_{\pm0.4}$ | $65.3_{\pm1.1}$ |
| | Correlation R. | $100.0_{\pm0.0}$ | $100.0_{\pm0.0}$ | $100.0_{\pm0.0}$ | $100.0_{\pm0.0}$ |
| | Uncertain+LR | $64.3_{\pm0.4}$ | $62.4_{\pm2.0}$ | $64.9_{\pm0.3}$ | $62.8_{\pm0.9}$ |
| | Uncertain+TabPFN | $\mathbf{57.5}_{\pm3.3}$ | $\mathbf{47.0}_{\pm8.6}$ | $\mathbf{61.1}_{\pm3.0}$ | $\mathbf{53.9}_{\pm6.1}$ |
| ACSMobility | Vanilla | $68.3_{\pm0.8}$ | $67.8_{\pm1.0}$ | $67.6_{\pm1.0}$ | $67.4_{\pm1.2}$ |
| | Group Balanced | $68.1_{\pm0.7}$ | $67.9_{\pm1.4}$ | $67.6_{\pm1.1}$ | $67.6_{\pm1.5}$ |
| | Correlation R. | $100.0_{\pm0.0}$ | $100.0_{\pm0.0}$ | $100.0_{\pm0.0}$ | $100.0_{\pm0.0}$ |
| | Uncertain+LR | $68.1_{\pm0.9}$ | $67.8_{\pm0.9}$ | $67.4_{\pm0.8}$ | $66.9_{\pm0.8}$ |
| | Uncertain+TabPFN | $\mathbf{68.1}_{\pm0.8}$ | $\mathbf{67.7}_{\pm1.0}$ | $\mathbf{67.2}_{\pm0.8}$ | $\mathbf{66.8}_{\pm0.9}$ |
| German Credit | Vanilla | $72.5_{\pm3.1}$ | $70.3_{\pm3.1}$ | $71.8_{\pm3.1}$ | $71.0_{\pm3.0}$ |
| | Group Balanced | $72.7_{\pm2.3}$ | $70.7_{\pm2.3}$ | $71.9_{\pm3.0}$ | $71.2_{\pm2.6}$ |
| | Correlation R. | $100.0_{\pm0.0}$ | $100.0_{\pm0.0}$ | $100.0_{\pm0.0}$ | $100.0_{\pm0.0}$ |
| | Uncertain+LR | $64.6_{\pm5.1}$ | $62.3_{\pm8.5}$ | $68.4_{\pm4.8}$ | $67.8_{\pm5.4}$ |
| | Uncertain+TabPFN | $\mathbf{60.4}_{\pm5.5}$ | $\mathbf{51.3}_{\pm26.5}$ | $\mathbf{63.8}_{\pm3.9}$ | $\mathbf{60.8}_{\pm10.9}$ |
| Diabetes | Vanilla | $80.2_{\pm0.1}$ | $89.0_{\pm0.1}$ | $80.4_{\pm0.1}$ | $89.0_{\pm0.1}$ |
| | Group Balanced | $\mathbf{66.2}_{\pm0.9}$ | $\mathbf{75.9}_{\pm0.8}$ | $\mathbf{65.1}_{\pm0.4}$ | $\mathbf{74.6}_{\pm0.4}$ |
| | Correlation R. | $100.0_{\pm0.0}$ | $100.0_{\pm0.0}$ | $100.0_{\pm0.0}$ | $100.0_{\pm0.0}$ |
| | Uncertain+LR | $70.4_{\pm20.7}$ | $77.1_{\pm26.8}$ | $80.3_{\pm0.1}$ | $89.0_{\pm0.1}$ |
| | Uncertain+TabPFN | $74.9_{\pm10.0}$ | $84.8_{\pm8.0}$ | $80.2_{\pm0.1}$ | $89.0_{\pm0.1}$ |
| CelebA | Vanilla | $84.7_{\pm0.2}$ | $83.2_{\pm0.3}$ | $85.0_{\pm0.2}$ | $83.2_{\pm0.3}$ |
| | Group Balanced | $84.6_{\pm0.2}$ | $83.2_{\pm0.3}$ | $84.9_{\pm0.2}$ | $83.3_{\pm0.3}$ |
| | Correlation R. | $100.0_{\pm0.0}$ | $100.0_{\pm0.0}$ | $100.0_{\pm0.0}$ | $100.0_{\pm0.0}$ |
| | Uncertain+LR | $\mathbf{72.4}_{\pm11.7}$ | $\mathbf{61.2}_{\pm21.7}$ | $84.9_{\pm0.2}$ | $83.1_{\pm0.3}$ |
| | Uncertain+TabPFN | $74.5_{\pm8.4}$ | $70.8_{\pm10.1}$ | $\mathbf{81.2}_{\pm7.2}$ | $\mathbf{77.2}_{\pm11.9}$ |

Table 1: **ICL prediction performance of sensitive attributes after applying different fairness interventions**. Smaller accuracy is better since it indicates how well the foundation model can reconstruct the sensitive attribute after the pre-processing fairness interventions. `Uncertain` methods yield the smallest accuracy, which justifies the improved fairness performance.

signals to predict the target variable and thereby amplifying unfairness. We provide further analysis in Section 4.5, showing that applying correlation removal only to the training set—while keeping the test set unchanged—improves fairness compared to the `Vanilla` baseline.

Finally, the `Uncertain` methods yield the lowest reconstruction accuracy across datasets. This suggests that the selected in-context samples carry little information about the sensitive attribute, limiting the model's ability to exploit it and thereby improving fairness.

### 4.2.3 Comparison of Foundation Models: TabICL vs. TabDPT vs. TabPFN

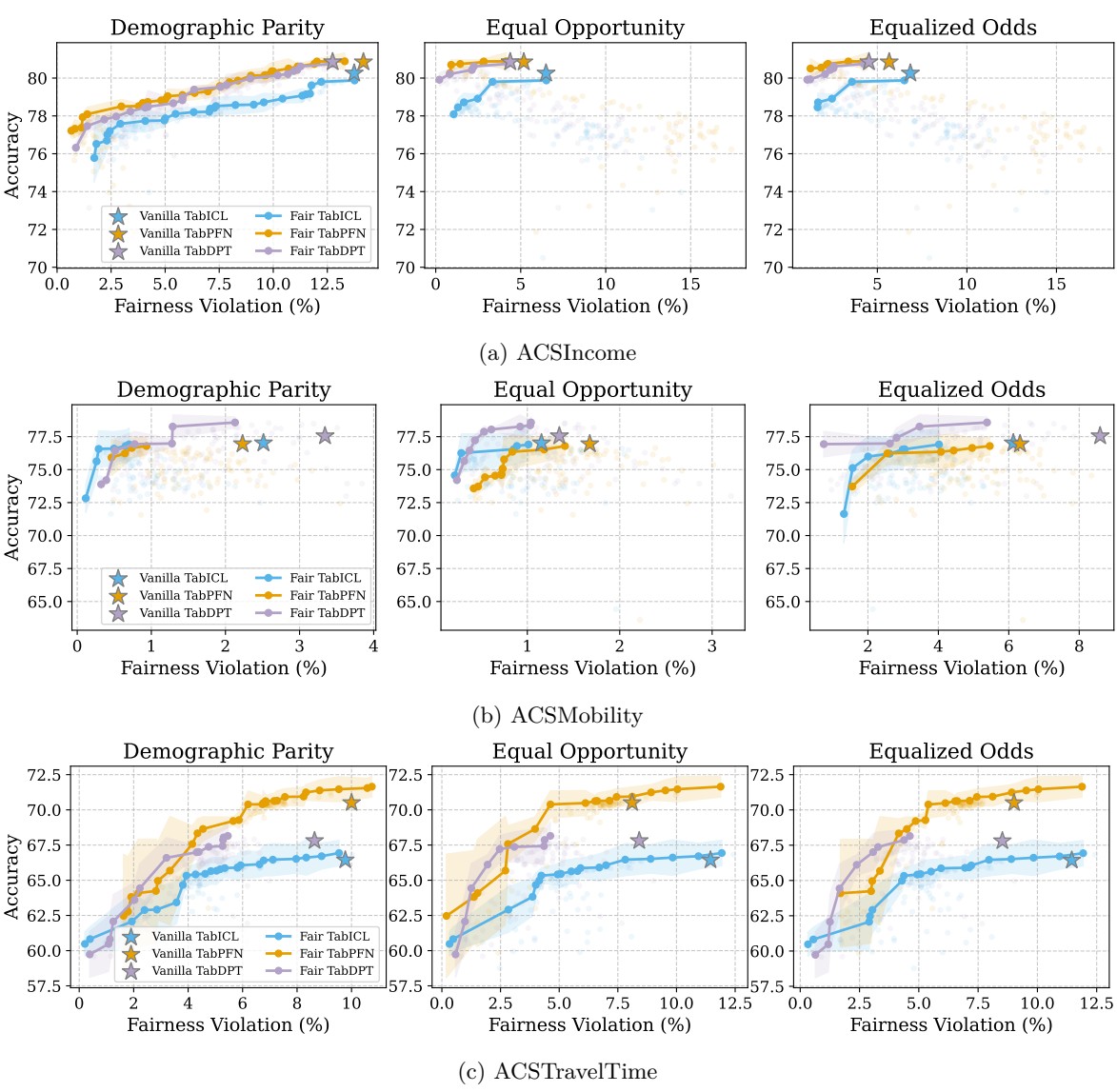

(a) ACSIncome

(b) ACSMobility

(c) ACSTravelTime

Figure 2: Comparing the fairness-utility tradeoffs of tabular foundation models (TabICL, TabDPT, and TabPFN) under uncertainty-based in-context sample selection (`Uncertain+TabPFN`) for different coverage ($\epsilon$ controlling the tradeoff). Results with other datasets can be found in the Appendix (Figure 5).

In the previous experiment, we compared the `Correlation Remover` and `Uncertain` methods in terms of their fairness–accuracy tradeoffs by varying their respective control parameters. The results showed that the `Uncertain` methods often achieved better Pareto-dominant points compared to both `Vanilla`

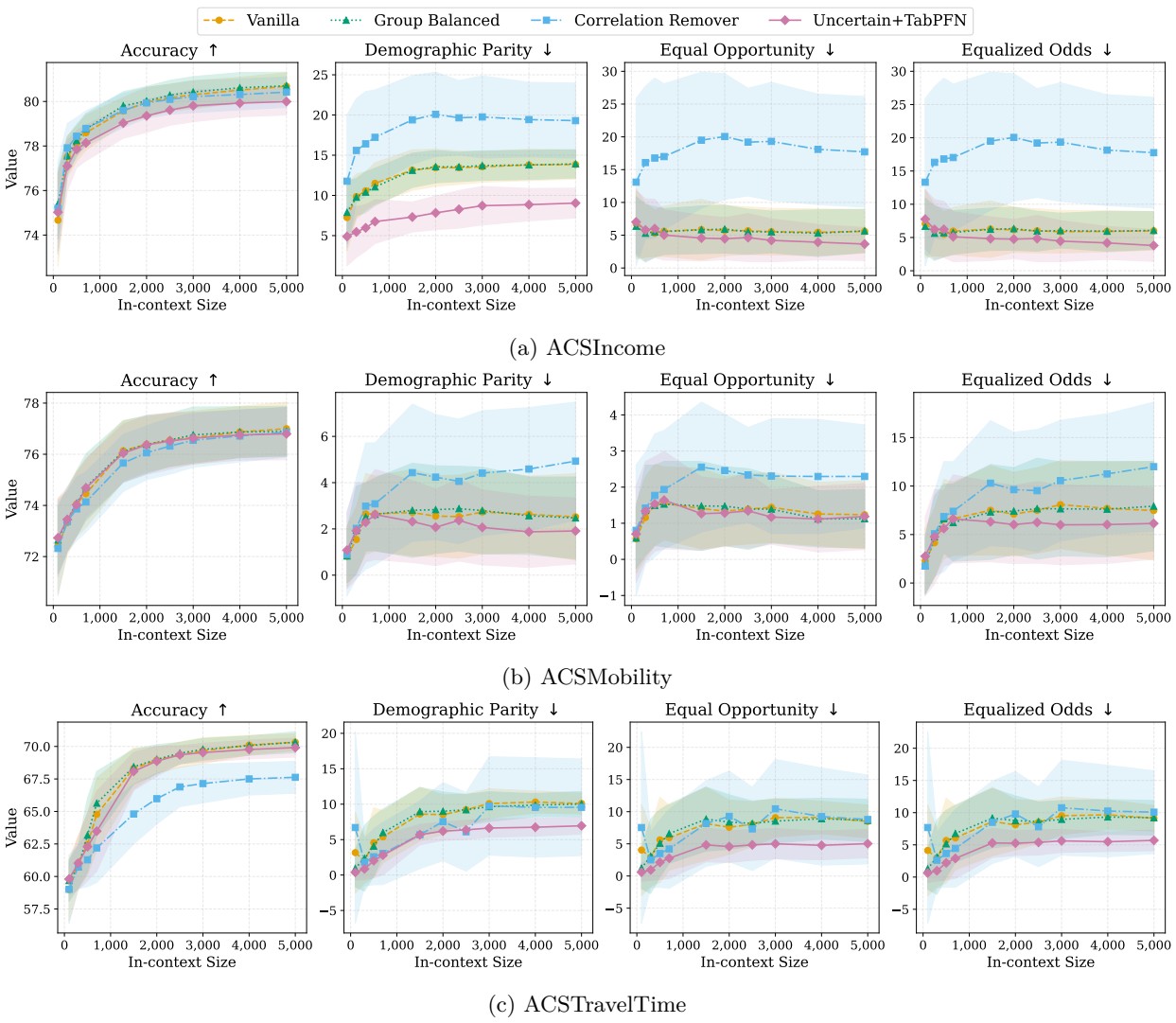

Figure 3: **Ablation on the in-context example set size**. Analyzing the impact of the in-context set size on the fairness and accuracy of ICL prediction with TabPFN.

ICL and `Correlation Remover`. Notably, using TabPFN for uncertainty estimation (`Uncertain+TabPFN`) outperformed the variant that relies on logistic regression (`Uncertain+LR`).

In this experiment, we extend the comparison to different foundation models by applying the same setup using our best-performing fairness intervention: `Uncertain+TabPFN`. Figures 2a and 2b show the fairness–accuracy Pareto fronts on the ACSIncome and ACSMobility datasets, respectively.

Without any fairness intervention (i.e., using the vanilla ICL approach), there is no clear winner among the foundation models in terms of fairness; the results vary across datasets. While one might expect TabDPT—pretrained on real-world data—to encode more real-world biases and thus exhibit poorer fairness performance, this is not consistently observed. In fact, TabDPT often performs comparably to or better than foundation models pretrained on synthetic data (R3).

Under fairness intervention with `Uncertain+TabPFN`, all three foundation models exhibit similar fairness performance, demonstrating that the `Uncertain` method can consistently control the fairness–accuracy trade-off regardless of the underlying foundation model. However, both TabPFN and TabDPT consistently achieve higher accuracy across datasets (R3). These findings are consistent with previous studies: the

benchmark Erickson et al. (2025) reported strong performance for TabPFN (Qu et al., 2025), while the authors of TabDPT highlighted its competitive performance relative to TabPFN. We observe similar patterns across additional datasets, and results are provided in the Appendix (see Figure 5).

## 4.3 Ablation on Impact In-context Sample Size

In all previous experiments, we used the full training set as the in-context example set whenever possible. For example, the current version of TabPFN is limited to handling a maximum of 10,000 samples (Hollmann et al., 2025). To understand the impact of in-context set size, we conduct an ablation study by varying the number of in-context examples across the range [100,300,500,700,1500,2000,2500,3000,4000,5000], while keeping the evaluation setup identical as described above.

Figure 3 shows that accuracy increases substantially with larger in-context sets, particularly in the lower range. However, unfairness shows a slight increase initially and then stabilizes once the in-context set exceeds approximately 700 examples. Across all in-context sizes and datasets, the `Correlation Remover` method consistently exhibits the highest level of unfairness, reinforcing earlier observations of its bias amplification. In contrast, the `Uncertain+TabPFN` method consistently achieves the lowest fairness violation, demonstrating robustness to the size of the in-context set. This indicates that even when only a subset of training data can be used, `Uncertain+TabPFN` remains effective at mitigating bias without overly compromising accuracy. We observe a similar trend when using TabICL model, and results are provided in Appendix 13.

In Appendix C, we perform a computational analysis comparing running and inference time of different fairness interventions and foundation models. This analysis shows that fairness interventions themselves are computationally inexpensive. The runtime is dominated by the choice of the tabular foundation model. TabPFN is the most efficient, benefiting from its compact architecture. The uncertainty-based variant with LR is fastest, while using it with TabPFN itself for uncertainty estimation adds moderate overhead..

## 4.4 Comparison with LLM-based ICL Methods

| Method | Foundation model | **Accuracy** | $\mathbf{\Delta}_{\text{DP}}$ | $\mathbf{\Delta}_{\text{EOP}}$ |
|---|---|---|---|---|
| Vanilla | TabPFN | $\mathbf{85.72_{\pm 0.3}}$ | $16.99_{\pm 1.0}$ | $\mathbf{8.80_{\pm 5.1}}$ |
| Vanilla | LLaMA-2-13B | $76.00_{\pm 1.1}$ | $\mathbf{14.00_{\pm 0.4}}$ | $11.00_{\pm 0.8}$ |
| Hu et al. (2024) | GPT-3.5-turbo | $77.93$ | $6.64$ | $10.9$ |
| Bhaila et al. (2024) | LLaMA-2-13B | $75.72_{\pm 1.6}$ | $8.00_{\pm 2.0}$ | $\mathbf{3.00_{\pm 3.0}}$ |
| Uncertain+TabPFN | TabPFN | $\mathbf{82.05_{\pm 1.0}}$ | $\mathbf{5.51_{\pm 1.4}}$ | $3.86_{\pm 1.4}$ |

Table 2: Comparison of accuracy and fairness metrics ($\Delta_{\text{DP}}$, $\Delta_{\text{EOP}}$) on the ADULT dataset.

Despite LLMs' in-context learning capabilities, they often lag behind classical machine learning models on large-scale tabular datasets (Hegselmann et al., 2023). To validate the effectiveness of our uncertainty-based demonstration selection method, we compare it against fairness-aware LLM-based fair ICL approaches proposed by Hu et al. (2024) and Bhaila et al. (2024) on the widely studied ADULT dataset (Asuncion et al., 2007). Table 2 presents the results. Vanilla TabPFN achieves an accuracy of 85.72%, significantly outperforming the LLaMA-2-13B baseline, which reaches only 76.00%. This 10-point gap underscores the performance advantage of specialized models for tabular data, consistent with prior findings that LLMs struggle on such inputs when used naively (Hegselmann et al., 2023).

While these comparisons rely on LLMs that are not the most recent generation, and performance may have improved with newer models, the results nonetheless highlight the continued strength of tabular foundation models like TabPFN in this setting. Moreover, when fairness interventions are applied, our uncertainty-based selection strategy consistently preserves high predictive performance while achieving comparable or better fairness outcomes than the LLM-based fair ICL approaches—demonstrating that principled sample selection can improve fairness without sacrificing accuracy.

| Dataset | Fairness Intervention | TabPFN | | TabICL | |
|---------|----------------------|--------|---|--------|---|
| | | Accuracy ↓ | F1 Score ↓ | Accuracy ↓ | F1 Score ↓ |
| ACSIncome | None | $77.2_{\pm 0.5}$ | $78.4_{\pm 0.3}$ | $75.0_{\pm 0.5}$ | $76.18_{\pm 0.3}$ |
| | Correlation R. (S1) | $100.0_{\pm 0.0}$ | $100.0_{\pm 0.0}$ | $99.9_{\pm 0.0}$ | $99.93_{\pm 0.0}$ |
| | Correlation R. (S2) | $\mathbf{53.8}_{\pm 0.4}$ | $\mathbf{67.0}_{\pm 0.9}$ | $\mathbf{52.9}_{\pm 0.3}$ | $\mathbf{68.9}_{\pm 0.3}$ |
| ACSTravelTime | None | $75.9_{\pm 0.4}$ | $77.5_{\pm 0.5}$ | $72.8_{\pm 0.4}$ | $73.75_{\pm 0.5}$ |
| | Correlation R. (S1) | $100.0_{\pm 0.0}$ | $100.0_{\pm 0.0}$ | $100.0_{\pm 0.0}$ | $100.00_{\pm 0.0}$ |
| | Correlation R. (S2) | $\mathbf{53.6}_{\pm 1.4}$ | $\mathbf{54.2}_{\pm 14.5}$ | $\mathbf{51.8}_{\pm 1.4}$ | $\mathbf{66.4}_{\pm 3.4}$ |
| ACSPublicCoverage | None | $91.4_{\pm 0.2}$ | $88.9_{\pm 0.2}$ | $91.5_{\pm 0.1}$ | $89.23_{\pm 0.2}$ |
| | Correlation R. (S1) | $100.0_{\pm 0.0}$ | $100.0_{\pm 0.0}$ | $100.0_{\pm 0.0}$ | $100.00_{\pm 0.0}$ |
| | Correlation R. (S2) | $\mathbf{57.8}_{\pm 0.4}$ | $\mathbf{0.1}_{\pm 0.1}$ | $\mathbf{56.5}_{\pm 1.0}$ | $\mathbf{0.1}_{\pm 0.1}$ |
| ACSEmployment | None | $64.0_{\pm 0.4}$ | $62.0_{\pm 1.8}$ | $65.0_{\pm 0.3}$ | $62.23_{\pm 1.4}$ |
| | Correlation R. (S1) | $100.0_{\pm 0.0}$ | $100.0_{\pm 0.0}$ | $100.0_{\pm 0.0}$ | $100.00_{\pm 0.0}$ |
| | Correlation R. (S2) | $\mathbf{52.6}_{\pm 0.6}$ | $\mathbf{27.7}_{\pm 4.4}$ | $\mathbf{53.7}_{\pm 5.1}$ | $\mathbf{53.0}_{\pm 10.6}$ |
| ACSMobility | None | $68.3_{\pm 0.8}$ | $67.8_{\pm 1.0}$ | $67.6_{\pm 1.0}$ | $67.40_{\pm 1.2}$ |
| | Correlation R. (S1) | $100.0_{\pm 0.0}$ | $100.0_{\pm 0.0}$ | $100.0_{\pm 0.0}$ | $100.00_{\pm 0.0}$ |
| | Correlation R. (S2) | $\mathbf{49.2}_{\pm 1.5}$ | $\mathbf{49.2}_{\pm 13.2}$ | $\mathbf{49.2}_{\pm 0.8}$ | $\mathbf{40.2}_{\pm 31.2}$ |
| Diabetes | None | $80.2_{\pm 0.1}$ | $89.0_{\pm 0.1}$ | $80.4_{\pm 0.1}$ | $89.04_{\pm 0.1}$ |
| | Correlation R. (S1) | $100.0_{\pm 0.0}$ | $100.0_{\pm 0.0}$ | $100.0_{\pm 0.0}$ | $100.00_{\pm 0.0}$ |
| | Correlation R. (S2) | $\mathbf{67.8}_{\pm 13.8}$ | $\mathbf{78.9}_{\pm 12.2}$ | $\mathbf{79.3}_{\pm 0.9}$ | $\mathbf{88.4}_{\pm 0.6}$ |
| German Credit | None | $72.5_{\pm 3.1}$ | $70.3_{\pm 3.1}$ | $71.8_{\pm 3.1}$ | $71.02_{\pm 3.0}$ |
| | Correlation R. (S1) | $100.0_{\pm 0.0}$ | $100.0_{\pm 0.0}$ | $100.0_{\pm 0.0}$ | $100.00_{\pm 0.0}$ |
| | Correlation R. (S2) | $\mathbf{37.8}_{\pm 6.7}$ | $\mathbf{44.0}_{\pm 14.5}$ | $\mathbf{46.3}_{\pm 4.2}$ | $\mathbf{38.8}_{\pm 14.6}$ |
| CelebA | None | $84.7_{\pm 0.2}$ | $83.2_{\pm 0.3}$ | $85.0_{\pm 0.2}$ | $83.16_{\pm 0.3}$ |
| | Correlation R. (S1) | $100.0_{\pm 0.0}$ | $100.0_{\pm 0.0}$ | $100.0_{\pm 0.0}$ | $100.00_{\pm 0.0}$ |
| | Correlation R. (S2) | $\mathbf{54.8}_{\pm 0.3}$ | $\mathbf{1.0}_{\pm 0.8}$ | $\mathbf{45.2}_{\pm 0.2}$ | $\mathbf{62.2}_{\pm 0.2}$ |

Table 3: Accuracy ICL prediction of sensitive attribute after applying `Correlation Remover` on the training and testing datasets (S1) or only to train dataset (S2). Applying the transformation only to the train dataset significantly reduces the accuracy of predicting the sensitive attribute.

## 4.5 On the Failure of Correlation Remover

In our previous experiments, we observed that applying the CR to both the training and test data can exacerbate unfairness in ICL predictions. We hypothesized that the foundation model infers the sensitive attribute from the linear transformation applied to each non-sensitive feature, which inadvertently leaks sensitive information and increases unfairness. To validate this hypothesis, we conducted correlation removal prediction of the sensitive attribute after applying fairness interventions. As shown in Table 1, the ICL prediction of the sensitive attribute achieves 100% accuracy following the application of CR. This indicates that the foundation model continues to rely heavily on the sensitive attribute even after correlation removal is performed.

For further verification, we tested a variant of correlation removal where the feature transformation (see Eq. 10 in Appendix) is applied only to the training data, leaving the test data unchanged (referred to as variant S2). Table 3 shows that this variant significantly reduces the ICL prediction accuracy of the sensitive attribute. This demonstrates that the foundation model exploits the transformation applied to the test set in the original correlation removal method (variant S1) as a proxy to fully reconstruct sensitive attributes.

This observation aligns with prior discussions by Aïvodji et al. (2021) regarding scenarios where data transformations reduce correlation with sensitive attributes. Besides the classical correlation removal setting

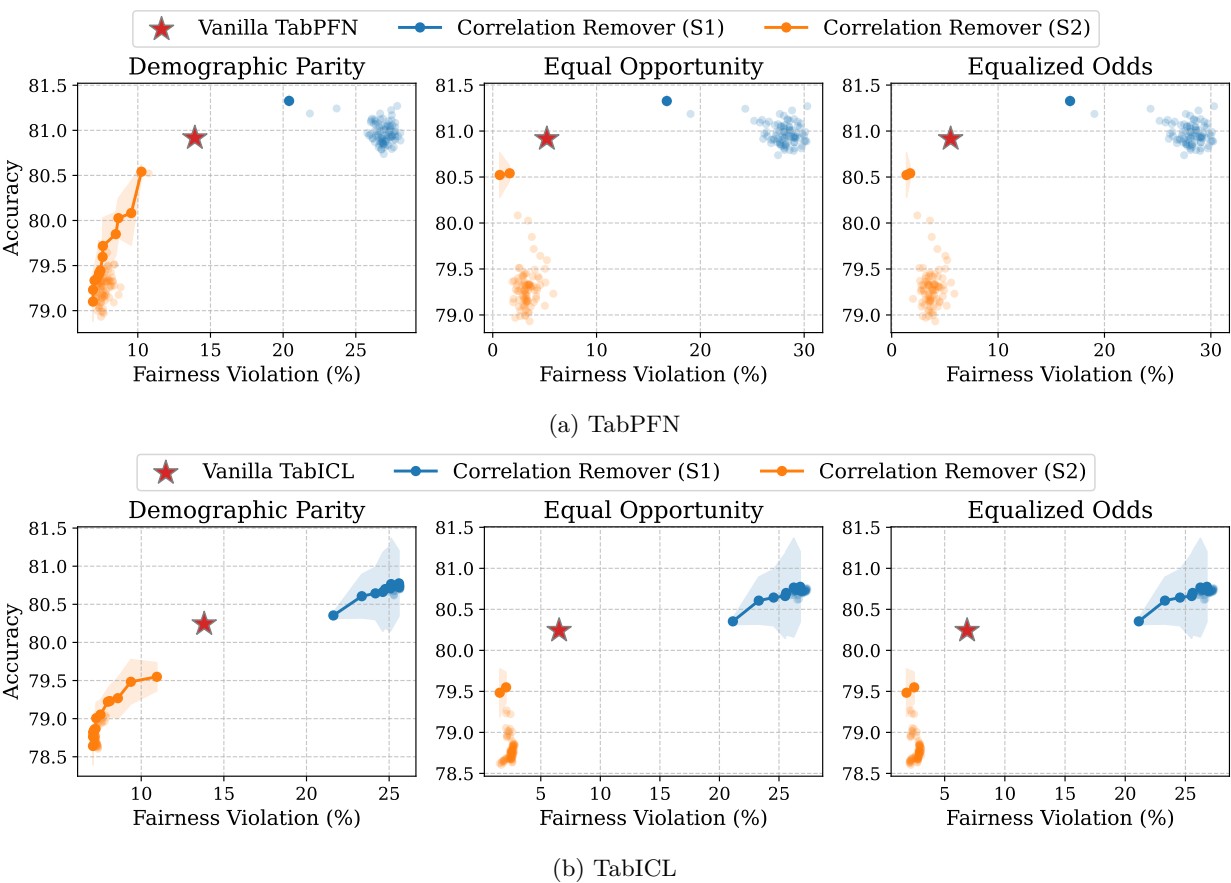

(a) TabPFN

(b) TabICL

Figure 4: **Evaluating variants of the correlation remover on the ACSIncome dataset with TabPFN and TabICL**. Applying correlation remover to the training and testing data exacerbates unfairness, while applying the transformation only to the training set improves fairness.

(S1), where transformations are applied to both training and test sets, we evaluated the fairness–accuracy trade-off when applying the transformation only to the training set (S2).

As illustrated in Figure 4, variant S2 of CR substantially improves fairness compared to S1. This confirms the foundation models' capacity to reconstruct sensitive information when the test set is transformed, leading to bias amplification. And the drop in reconstruction accuracy of the sensitive attribute directly supports the observed improvement in fairness when using variant S2

Based on these findings, we recommend practitioners apply the S2 variant of Correlation Remover—transforming only the training data—when using correlation removal in in-context learning frameworks, to avoid sensitive attribute leakage and mitigate bias amplification during testing.

## 5   Conclusion and future works

In this study, we proposed and analyzed the effectiveness of three preprocessing methods to enhance the fairness of in-context learning (ICL) predictions. Our empirical results, performed on eight fairness benchmarks, posit the uncertainty-based in-context selection method as a strong baseline for improving the fairness of tabular ICL. The key advantages of this method are threefold: (1) it does not require fine-tuning or retraining the foundation model to enforce the desired fairness metrics; (2) it can consistently improve three widely used group fairness metrics; (3) it offers a parameter to control the fairness-utility tradeoff. To our knowledge, this is the first work that explores pre-processing fairness intervention on tabular foundation models. We hope

this work will trigger more investigations into fair tabular ICL, since in-context learning as a new learning paradigm will be increasingly adopted into decision-making tools. Interesting future research directions include investigating in-processing and post-processing methods and analyzing the effect of distribution shift between in-context and test examples on fairness and accuracy.

## Ethics Statement

This paper explores ways to reduce unfairness in tabular foundation models, emphasizing fair treatment for various groups. We recognize the significance of fairness in machine learning, especially regarding sensitive attributes like race, gender, and socio-economic status. Our research seeks to uncover and tackle potential biases in these models, thereby enhancing transparency, accountability, and inclusivity. While the proposed method uses a sensitive attributes predictor, which could be unlawful in some countries, we emphasized that predicted sensitive values are not used either for training or measuring unfairness. We use the attribute classifier only to quantify uncertainty, and emphasize that this method should not be used for any purpose other than bias measuring or mitigation.

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

## Appendix

## A  Datasets

We experiment on tasks from the recently proposed folktables (Ding et al., 2022), which contains data extracted from the American Community Survey (ACS) Public Use Microdata Sample (PUMS) (Ding et al., 2022). More specifically, we experiment with the following ACS PUMS tasks:

- **ACSIncome**: The task involves predicting whether an individual's income exceeds $50,000. The dataset is filtered to include only individuals over the age of 16 who reported working at least 1 hour per week during the past year and earned a minimum of $100.

- **ACSMobility**: This task involves predicting whether an individual had the same residential address one year ago. The dataset is filtered to include individuals aged between 18 and 35. This filtering increases the difficulty of the task, as more than 90% of the general population tends to stay at the same address year-to-year.

- **ACSTravelTime**: This task predicts whether an individual has a commute longer than 20 minutes. The dataset is filtered to include only employed individuals above the age of 16. The 20-minute threshold corresponds to the median commute time in the US, according to the 2018 ACS PUMS data.

- **ACSEmployment**: The objective is to predict whether an individual is employed, using a dataset filtered to include individuals aged between 16 and 90.

- **ACSPublicCoverage**: The goal is to predict whether an individual has public health insurance. The dataset is filtered to include individuals under 65 years of age and those with an income below $30,000, focusing on low-income individuals who are ineligible for Medicare.

These tasks were selected to reflect a range of real-world predictive challenges with fairness concerns. A limitation of the ACS PUMS datasets is that they are US-centric; we diversify the experimental setup by including other tasks and datasets. Specifically, we also evaluate on the following tabular datasets and tasks:

- **Diabetes** (Gardner et al., 2023): The diabetes prediction task uses features related to physical health, lifestyle factors, and chronic conditions, derived from the BRFSS questionnaires. Demographic attributes like race, sex, state, and income are also included. The target is a binary indicator of whether the respondent has ever been diagnosed with diabetes.

- **German Credit** (Frank, 2010): The German Credit dataset contains 20 attributes of 1,000 individuals. We create the task of classifying people according to whether they have a good or bad credit risk using age (over or below 25 years old) as the sensitive attribute.

- **CelebA** (Liu et al., 2018): The dataset contains 202,599 samples described with 40 facial attributes of human annotated images. We create the task of predicting *attractiveness* with facial attributes using gender as the sensitive attribute (Kenfack et al., 2024b). Note that we do not train the model with images and consider this task to diversify the experimental tasks.

## B  Background

### B.1  Fairness Metrics

In this work, we focus on group fairness notions that measure the performance disparity across different demographic groups. More specifically, we consider the following three widely used group fairness metrics:

Table 4: Summary of datasets used in our experiments. For each dataset, we report the number of features (including the sensitive attribute), the number of samples available, and the sensitive attribute used for fairness evaluation.

| Dataset | # Features | # Samples | Sensitive Feature | Prediction Task |
|---|---|---|---|---|
| ACSIncome | 10 | 22,268 | Gender | Income $\geq$ \$50,000 |
| ACSEmployment | 16 | 47,777 | Gender | Employment status |
| ACSTravelTime | 16 | 19,492 | Gender | Commute time over 20 minutes |
| ACSMobility | 21 | 8,625 | Gender | Residential mobility |
| ACSPublicCoverage | 19 | 18,525 | Gender | Public health insurance coverage |
| CelebA | 39 | 202,599 | Gender | Attractiveness |
| Diabetes | 183 | 38,575 | Race | Prior diabetes diagnose |
| German | 58 | 990 | Age | Credit risk |

- **Demographic parity (DP)**: DP enforces equal positive outcome rate for different groups (Dwork et al., 2012) and is defined as follows:

$$P(f(X) = 1|S = s) = P(f(X) = 1) \tag{1}$$

- **Equalized Odds (EOD)**: EOdds is satisfied when the model makes correct and incorrect predictions at the same rate for different demographic groups (Hardt et al., 2016). The metric enforces equal true positive and false positive rates across groups and is measured as follows;

$$P(f(X) = 1|S = 0, Y = y) = P(f(X) = 1|S = 1, Y = y), \ \forall y \in \{0, 1\} \tag{2}$$

- **Equalized Opportunity (EOP)**: In some settings, one can care more about assessing unfairness when the model makes correct predictions. EOP enforces equal true positive rates across groups, i.e., we only consider $y = 1$ in Eq. 2, i.e.,

$$P(f(X) = 1|S = 0, Y = 1) = P(f(X) = 1|S = 1, Y = 1) \tag{3}$$

Empirically, we measure each fairness considered, i.e., Demographic Parity ($\Delta$DP), Equal Opportunity ($\Delta$EOP), and Equalized Odds ($\Delta$EOD) as follows.

$$\Delta\text{DP} = \left| \mathop{\mathbb{E}}_{x|A=0}[\mathbb{I}\{f(x) = 1\}] - \mathop{\mathbb{E}}_{x|A=1}[\mathbb{I}\{f(x) = 1\}] \right| \tag{4}$$

Where $\mathbb{I}(\cdot)$ is the indicator function.

$$\Delta\text{EOD} = \alpha_0 + \alpha_1 \tag{5}$$

$$\Delta\text{EOP} = \alpha_1 \tag{6}$$

Where $\alpha_0$ and $\alpha_1$ measure the difference between the false positive and the true positive rates across groups, respectively, and are empirically measured as follows.

Where $\alpha_0$ and $\alpha_1$ measure the difference between the false positive and the true positive rates across groups, respectively, and are empirically measured as follows.

$$\alpha_j = \left| \mathop{\mathbb{E}}_{x|A=0,Y=j}[\mathbb{I}\{f(x)=1\}] - \mathop{\mathbb{E}}_{x|A=1,Y=j}[\mathbb{I}\{f(x)=1\}] \right| \quad j \in \{0,1\} \tag{7}$$

Since the disparities can be below 0.1 on some datasets, we scaled the fairness values reported throughout the paper by 100 to make it easier to read.

## B.2 Correlation Remover

The `Correlation Remover` (Feldman et al., 2015) is a preprocessing technique designed to eliminate linear correlations between sensitive attributes and non-sensitive features in a dataset. This method is particularly useful in mitigating biases that may arise due to such correlations, especially when employing linear models.

Considering a classification task with the given training data $\mathcal{D} = \{(x_i, y_i, s_i)\}_{i=1}^n$ where $x_i$ is an input feature vector, $y_i$ is the corresponding class label, and $s_i$ the corresponding demographic group.

To apply `Correlation Remover`, we assume the training data is formulated as follows:

- $\mathbf{X} \in \mathbb{R}^{n \times d}$ represents the training data matrix containing sensitive and non-sensitive features.

- $\mathbf{S} \in \mathbb{R}^{n \times m_s}$ a matrix of the sensitive features. For simplicity, we assumed in this work $m_s = 1$, which corresponds to a single binary sensitive attribute.

- $\mathbf{Z} \in \mathbb{R}^{n \times m_z}$ a matrix of non-senstive features such that $X = [S\ Z]$

The goal of `Correlation Remover` is to transform $Z$ into $Z^*$ such that $Z^*$ is uncorrelated with $S$, while retaining as much information from the original $Z$ as possible.

For each non-sensitive feature vector $\mathbf{z}^j \in \mathbb{R}^n$ (the $j$-th column of $\mathbf{Z}$), the algorithm solves the following least squares problem:

$$\min_{\mathbf{w}_j} \left\| \mathbf{z}^j - (\mathbf{S} - \mathbf{1}_n \bar{\mathbf{s}}^\top) \mathbf{w}_j \right\|_2^2 \tag{8}$$

where:

- $\bar{\mathbf{s}} = [\bar{\mathbf{s}}_1, \bar{\mathbf{s}}_2, \ldots, \bar{\mathbf{s}}_{m_s}]$ is the mean vector of the sensitive features, i.e., $\bar{\mathbf{s}}_j$ is the mean of $j$-th the sensitive feature.

- $\mathbf{1}_n$ is an $n$-dimensional column vector of ones.

- $\mathbf{w}_j \in \mathbb{R}^{m_z}$ is the weight vector that projects the centered sensitive features onto $\mathbf{z}_j$.

After computing the optimal weight vectors $\mathbf{w}_j^*$ for all $j \in \{1, \ldots, m_z\}$, they are assembled into a weight matrix $\mathbf{W}^* = [\mathbf{w}_1^*, \ldots, \mathbf{w}_{m_z}^*]$. The transformed non-sensitive features are then obtained by:

$$\mathbf{Z}^* = \mathbf{Z} - (\mathbf{S} - \mathbf{1}_n \bar{\mathbf{s}}^\top) \mathbf{W}^* \tag{9}$$

This operation effectively removes the linear correlations between $\mathbf{S}$ and $\mathbf{Z}$, resulting in $\mathbf{Z}^*$ that is uncorrelated with the sensitive features.

`Correlation Remover` introduces a tunable parameter $\alpha \in [0, 1]$ that controls the extent of correlation removal, i.e., (i) $\alpha = 1$ corresponds to full removal of linear correlations, thus best possible fairness; (ii) $\alpha = 0$ corresponds no transformation, the original data is used; (iii) $0 < \alpha < 1$ corresponds to partial removal, balancing between the original and transformed data, thus controlling the fairness accuracy tradeoff. More specifically, the final transformed dataset $\mathbf{X}'$ is computed as:

$$\mathbf{X}' = \alpha \mathbf{Z}^* + (1 - \alpha) \mathbf{Z} \tag{10}$$

Note that $\mathbf{X}'$ is derived using $\mathbf{Z}^*$, since $\mathbf{S}$ is dropped after transformation. The convex combination 10 allows practitioners to adjust the fairness accuracy tradeoff based on specific requirements of their application.

Equation 8 is optimized on the training dataset, and the optimal weight vectors $w_j^*$ are used to apply the transformation 10 on the test dataset.

### B.3 Uncertainty measure with conformal prediction

While any uncertainty measurement method could be used in our method, we employ *conformal prediction* due to its strong coverage guarantees. Specifically, we used Split Conformal Prediction (Angelopoulos et al., 2023), which is a distribution-free and model-agnostic method that provides **prediction sets** for classification tasks, ensuring a user-specified **coverage level** $1 - \epsilon$ with no assumptions beyond data exchangeability.

Given labeled data $\mathcal{D}_{\text{train}} = \{(x_i, s_i)\}_{i=1}^n$ with binary sensitive attribute $s_i \in \{0, 1\}$, we split the data into *proper training* set ( $\mathcal{D}_1$) and calibration set ($\mathcal{D}_2$). In the experiments, we did a 50%-50% split of the dataset with sensitive attribute to obtain $\mathcal{D}_1$ and $\mathcal{D}_2$.

We then train a probabilistic classifier $f : \mathcal{X} \to [0, 1]$, on $\mathcal{D}_1$, yielding predictions:

$$\hat{p}_i = f(x_i) = \mathbb{P}(S = 1 \mid X = x_i). \tag{11}$$

In our setup, $f$ could be Logistic Regression (Uncertain+LR) or TabPFN (Uncertain+TabPFN). The starting point for conformal prediction is what is called a *nonconformity measure*, a real-valued function that measures how a prediction is different from any possible class label.

**Nonconformity Scores**   After training $f$ on the proper training set, we use the calibration dataset to compute the nonconformity scores, which measure how far the prediction is from the true label. More specifically, for each calibration point $(x_i, s_i) \in \mathcal{D}_2$, we considered the nonconformity score is defined as:

$$c_i = |s_i - \hat{p}_i|. \tag{12}$$

We then compute the quantile threshold $\tau$ based on the user-defined target coverage $\epsilon \in [0, 1]$, e.g., $\epsilon = 0.05$ mean 95% coverage. Specifically, $\tau$ is defined based on $1 - \epsilon$ quantile of the nonconformity scores.

$$\tau = \text{Quantile}_{1-\epsilon}\left(\{c_i\}_{i=1}^{n_{\text{cal}}}\right). \tag{13}$$

**Prediction Set**   The quantile threshold is used to build the prediction set of data points from the test set. More specifically, for a test sample $x_{\text{test}}$, we compute the prediction probability $\hat{p}_{\text{test}} = f(x_{\text{test}})$ and derive its prediction set as follows:

$$\Gamma(x_{\text{test}}) = \{s \in \{0, 1\} : |s - \hat{p}_{\text{test}}| \leq \tau\}. \tag{14}$$

When the prediction set only contains $\{0\}$ or $\{1\}$, then the prediction is confident with at least $1-\epsilon$ probability, while when the prediction set contains both labels, i.e., $\{0, 1\}$, the prediction is considered uncertain. Our uncertainty-based demonstration selection method uses only samples whose prediction set contains two values. The coverage guarantee of conformal prediction holds under the assumption that the calibration and test data are *exchangeable*. We use the open-source implementation of split conformal prediction provided by the MAPIE Python package.

**Example.**   Consider a simplified task consisting of two non-sensitive features ($f^1$ and $f^2$), one binary target ($y$), and a binary sensitive attribute ($s$).

To estimate the uncertainties, we first train a sensitive attribute classifier, using a fraction of the dataset (20% in our experiments). We train the sensitive attribute classifier (Logistic Regression or TabPFN) using $f^1$ and $f^2$ to predict $s$. A conformal predictor returns for a given (test) sample $x$ a prediction set $\Gamma(x)$ that contains the true sensitive attribute with probability of at least $1 - \epsilon$. For example, setting $\epsilon = 0.05$ means 95% of the data will contain the true label in their prediction sets. The size of the prediction set, therefore, provides information about the prediction uncertainty, i.e., $|\Gamma(x)| = 1$ means the prediction is confident with at least $1 - \epsilon$ probability, and $|\Gamma(x)| = 2$ means the prediction is uncertain. The intuition of our method is that using samples with uncertain predictions makes it challenging for the foundation model to infer and rely on the sensitive attributes to make predictions on the target.

We therefore include a training example $x$ in the context set if $|\Gamma(x)| = 2$ and filter it out otherwise. $\epsilon$ helps to control the fairness-utility tradeoff since a smaller $\epsilon$ corresponds to filtering out more demonstration examples with higher confidence sensitive attributes prediction, which can better improve fairness while impacting accuracy as the size of the in-context set is reduced.

## C   Runtime Comparison of Fairness Interventions

In addition to fairness–utility tradeoffs, we provide a detailed comparison of the computational cost of the fairness interventions across tabular foundation models. While prior sections focused on predictive performance, practical deployment also requires understanding efficiency, particularly in large-scale or resource-constrained settings.

**Setup.**   We measured wall-clock running time on an NVIDIA A100 GPU (20 GB memory). Each run includes preprocessing, fairness intervention, and in-context learning inference using the full training set and test set. Results are averaged over three seeds, with standard deviations reported. The largest dataset in our experiments, CelebA, was used to provide representative estimates of runtime under high load. Table 5 summarizes runtimes across methods and models. We make the following observations: (i) TabPFN is the most efficient (8–47s), benefiting from its compact architecture. The uncertainty-based variant with LR is fastest, while using TabPFN itself for uncertainty estimation adds moderate overhead. (ii) TabICL shows moderate cost (57–97s), with relatively small variance across interventions. (iii) TabDPT is substantially slower ( 595s), reflecting the higher computational demands of its transformer-based architecture.

Across models, uncertainty-based with Logistic Regression is consistently the fastest intervention, while correlation removal and group balancing introduce negligible overhead compared to vanilla inference. Recall TabPFN uses at most 10K samples, which reduces the context size compared to other models, thereby explaining the faster inference. Likewise, the uncertainty-based intervention reduces the context size compared to other interventions. Running time on the ACSIncome dataset (Table 6) confirms that fairness interventions themselves are computationally inexpensive; the runtime is dominated by the choice of tabular foundation model. For practitioners, TabPFN with uncertainty-based sampling (LR) offers the best balance of fairness gains and efficiency, while TabDPT provides stronger model capacity at substantially higher cost.

| Model | Vanilla | Group Balanced | Correlation Remover | Uncertain+LR | Uncertain+TabPFN |
|---|---|---|---|---|---|
| TabDPT | $596.04 \pm 69.12$ | $594.98 \pm 68.35$ | $596.83 \pm 70.10$ | $595.58 \pm 65.73$ | $316.86 \pm 50.17$ |
| TabICL | $69.50 \pm 0.75$ | $67.84 \pm 0.85$ | $70.11 \pm 1.26$ | $57.05 \pm 1.24$ | $96.65 \pm 1.06$ |
| TabPFN | $10.71 \pm 1.27$ | $10.84 \pm 1.16$ | $13.94 \pm 1.35$ | $8.45 \pm 0.82$ | $47.18 \pm 0.65$ |

Table 5: Average running time (s) $\pm$ standard deviation on CelebA dataset across fairness interventions and tabular foundation models.

| Model | Vanilla | Group Balanced | Correlation Remover | Uncertain+LR | Uncertain+TabPFN |
|---|---|---|---|---|---|
| TabDPT | $154.35 \pm 53.13$ | $153.23 \pm 53.06$ | $151.75 \pm 52.88$ | $155.09 \pm 53.41$ | $303.63 \pm 52.21$ |
| TabICL | $45.05 \pm 0.62$ | $45.23 \pm 0.04$ | $44.63 \pm 0.62$ | $45.09 \pm 0.29$ | $45.43 \pm 1.40$ |
| TabPFN | $1.85 \pm 0.87$ | $1.96 \pm 1.07$ | $3.22 \pm 0.32$ | $1.87 \pm 0.91$ | $3.65 \pm 0.76$ |

Table 6: Average inference time (s) $\pm$ standard deviation on ACSIncome dataset across fairness interventions and tabular foundation models.

## D   Supplementary resutls

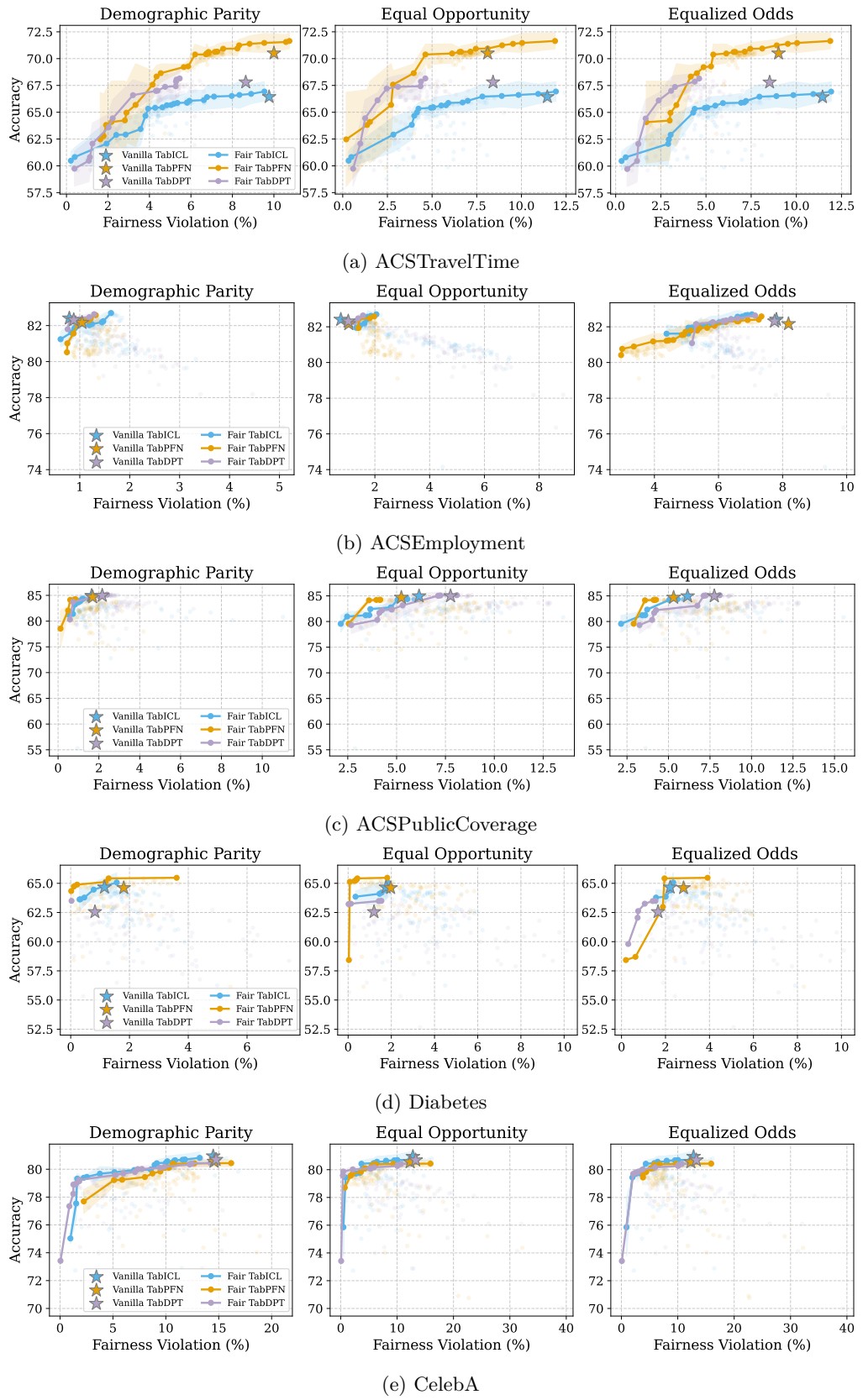

Figure 5: TabPFN vs. TabICL vs. TabDPT. Comparing the fairness-accuracy tradeoffs of tabular foundation models under uncertainty-based fairness interventions. TabPFN generally provides better fairness accuracy tradeoffs.

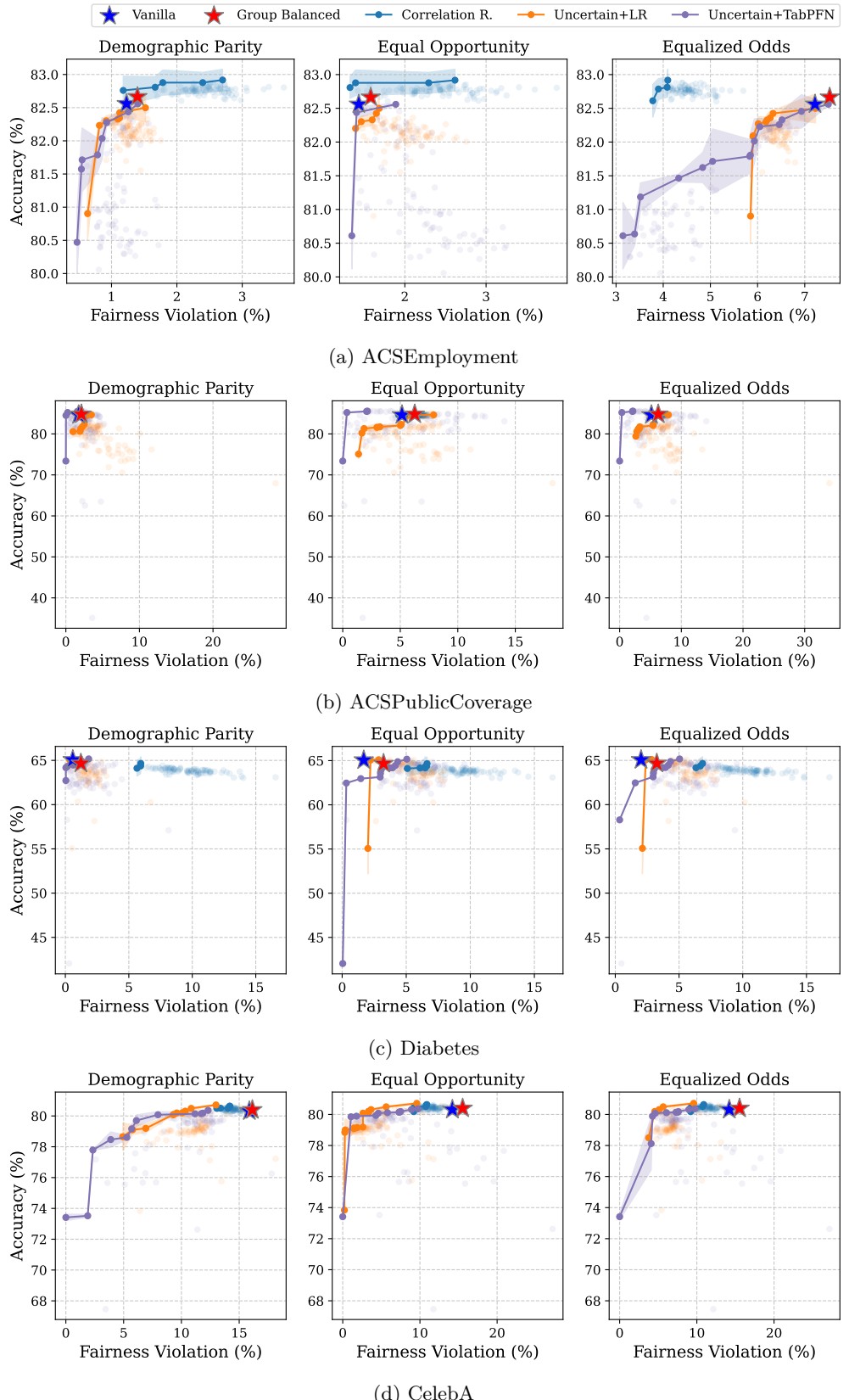

Figure 6: Fairness-accuracy Pareto-front of different fairness interventions with TabPFN.

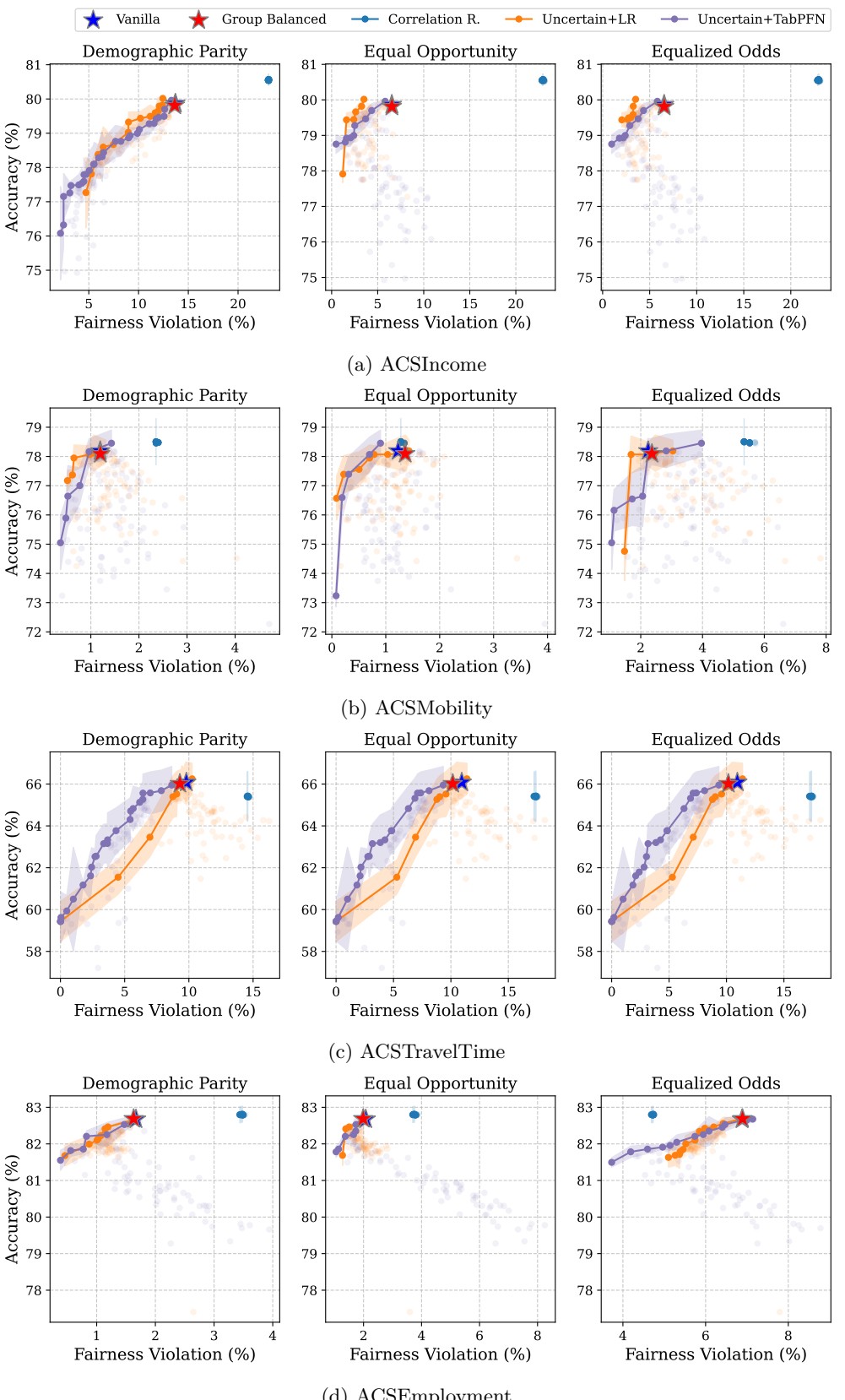

Figure 7: Fairness-accuracy tradeoffs on the ACS datasets with TabICL as foundation model
.

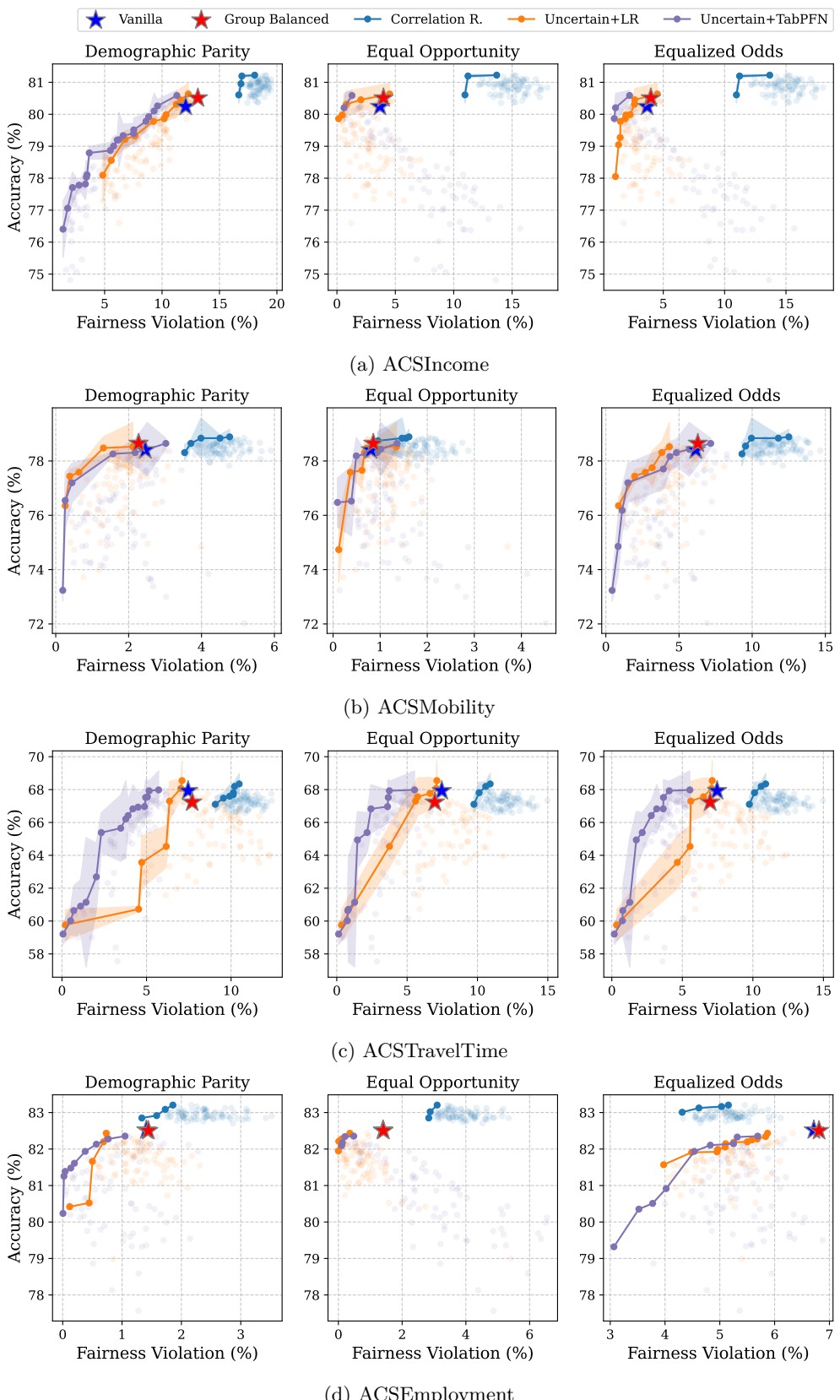

Figure 8: Fairness-accuracy tradeoffs on the ACS datasets with TabDPT as foundation model
.

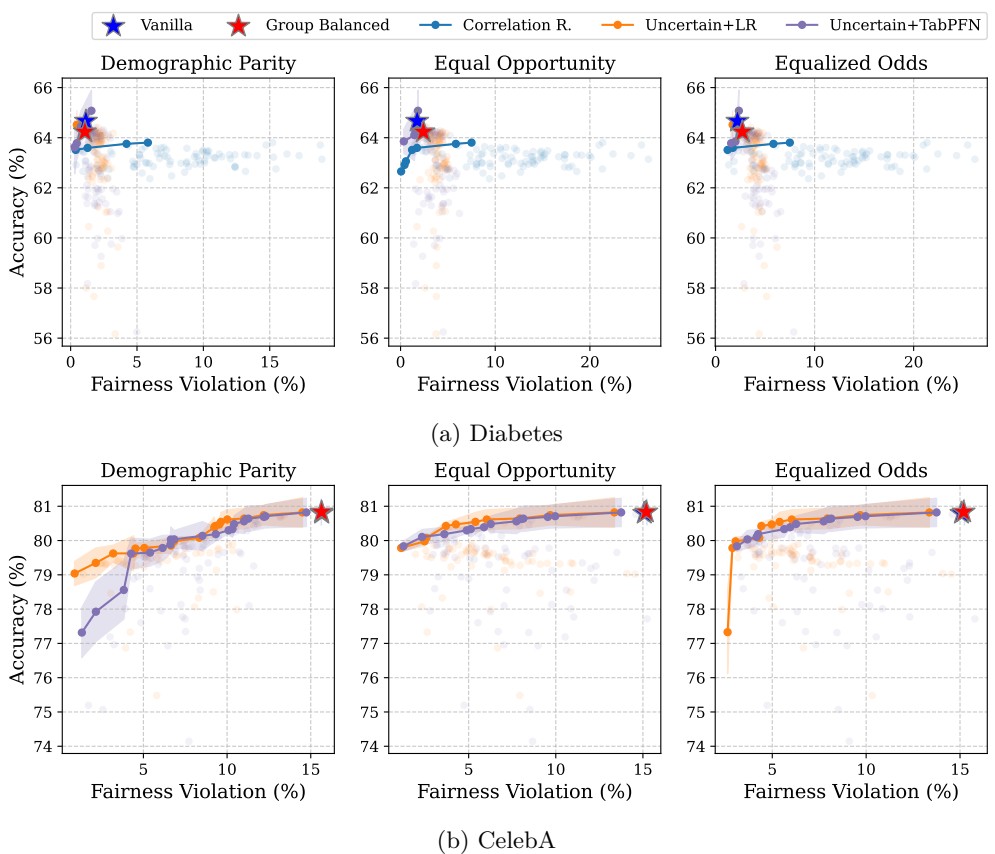

Figure 9: Fairness-accuracy tradeoffs on the Diabetes and CelebA using TabICL as foundation model
.

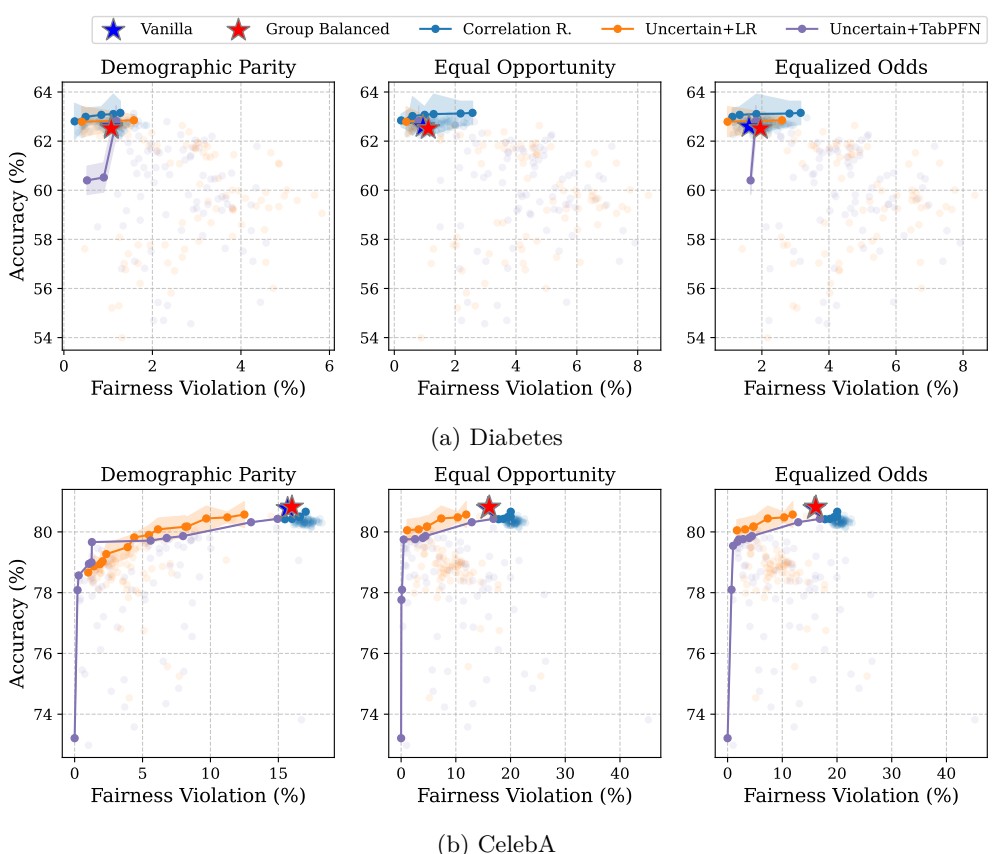

Figure 10: Fairness-accuracy tradeoffs on the Diabetes and CelebA using TabDPT as foundation model
.

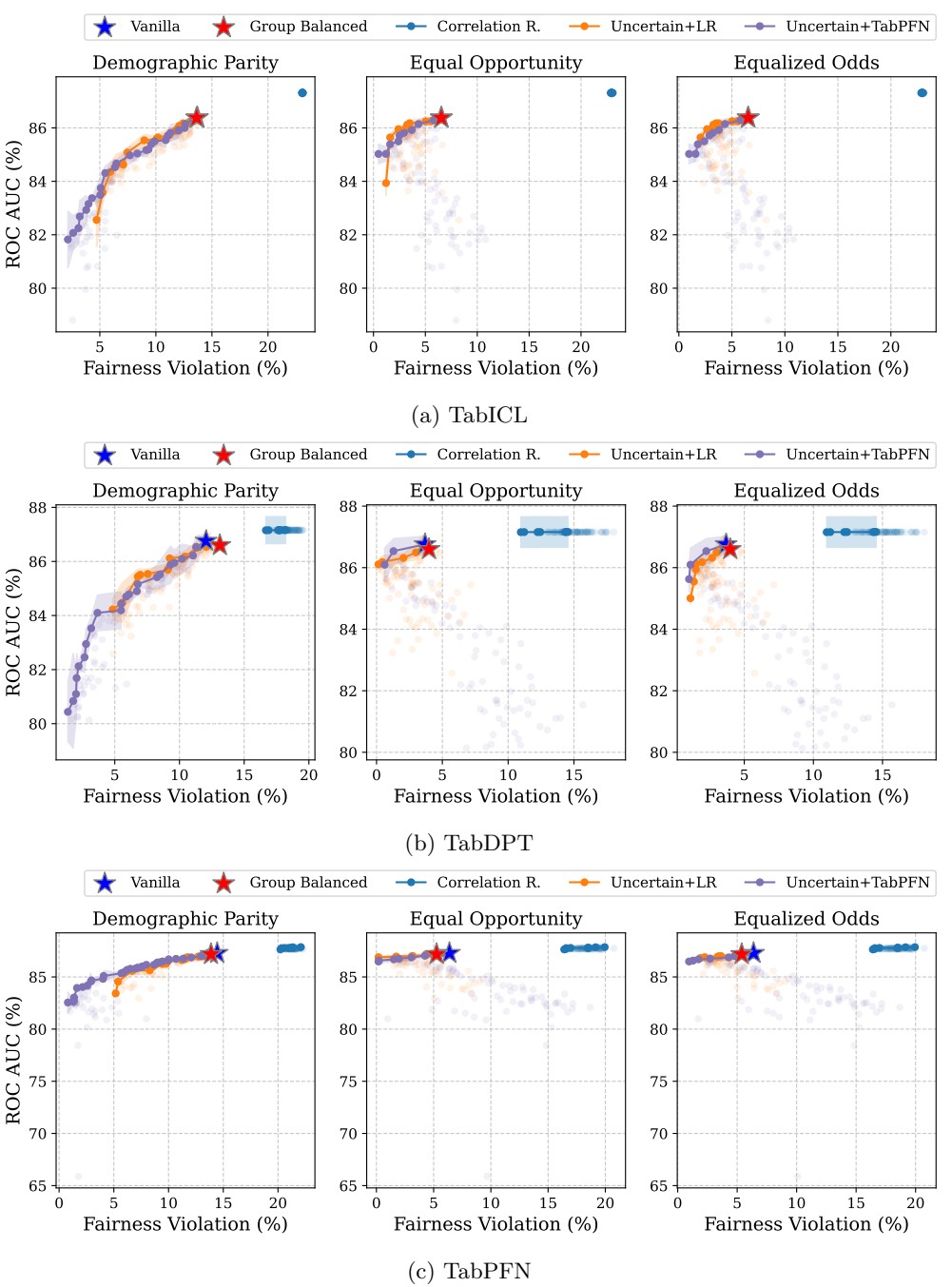

Figure 11: Fairness-utility tradeoff on the ACSIncome with utility measured by ROC-AUC.
.

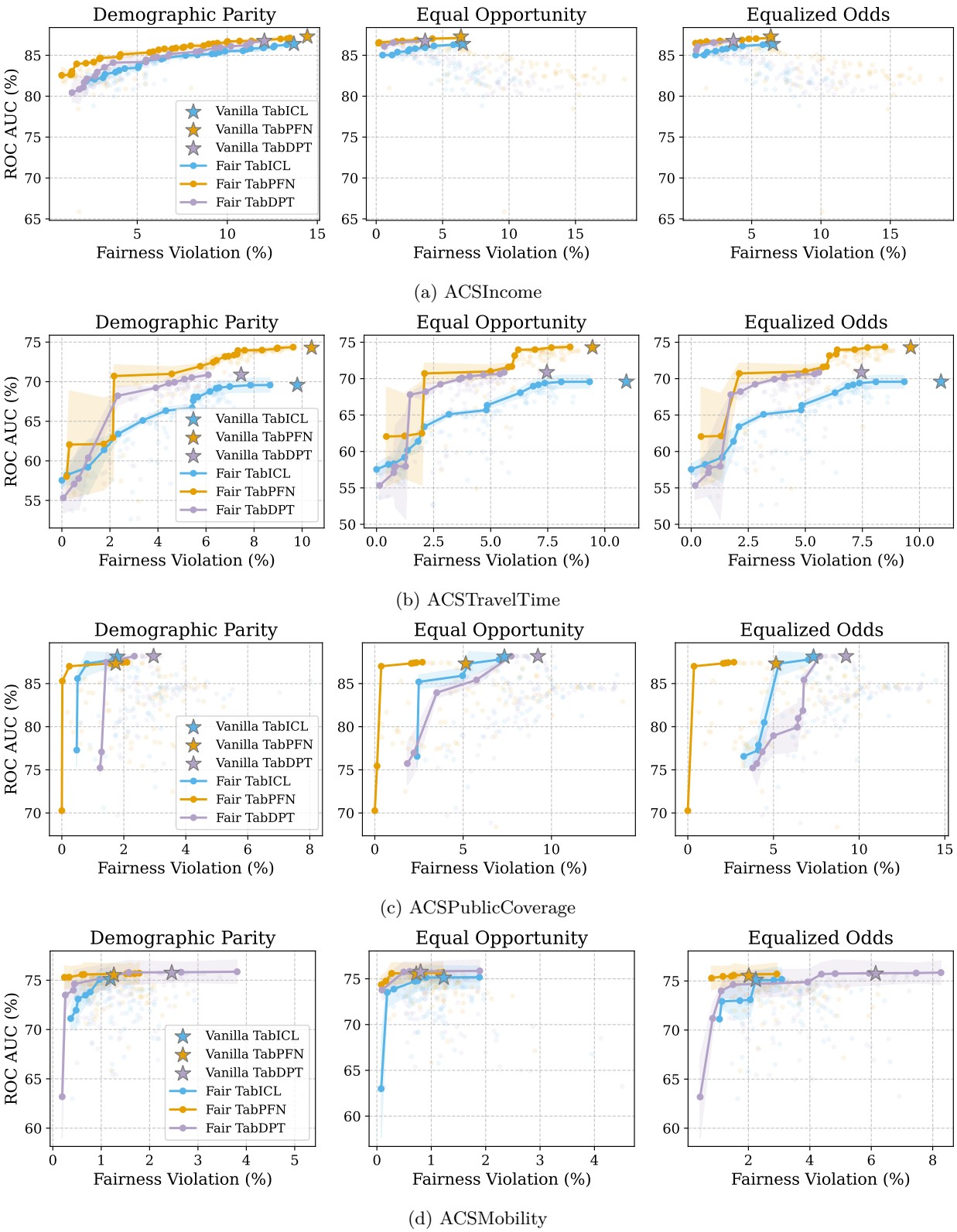

Figure 12: Comparing the fairness-utility (ROC AUC) tradeoff of different foundation models under `Uncertain+TabPFN` fairness intervention. Measuring the utility with the ROC AUC score shows a similar trend with accuracy.

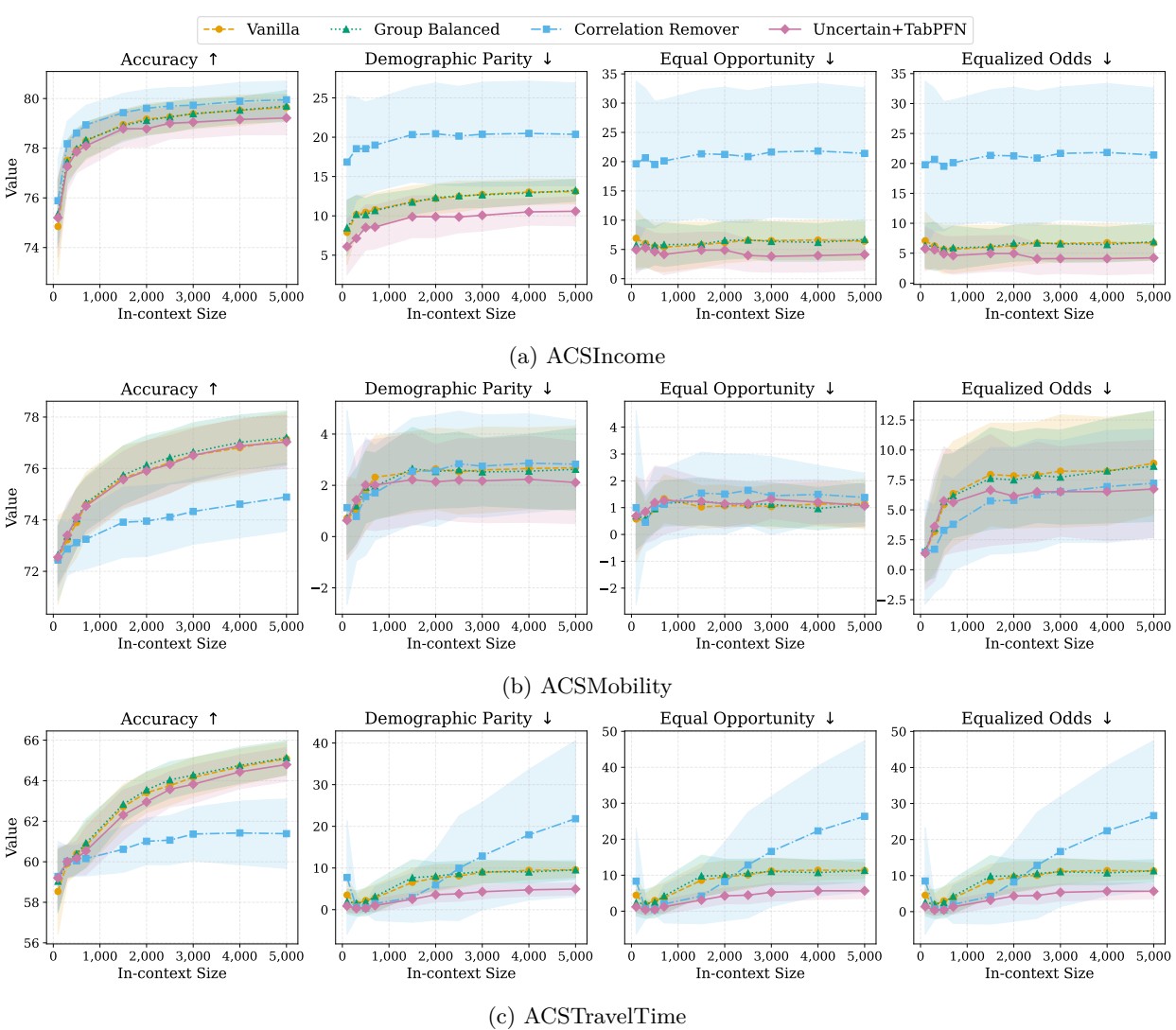

Figure 13: Ablation on the in-context example set size with TabICL.

