# OpenReview forum: "Towards Fair In-Context Learning with Tabular Foundation Models"
_TMLR — Accepted by TMLR_

### Review · Reviewer_qoEa · 2025-09-12

**Summary Of Contributions:**

The work focus on fairness in machine learning for tabular data. In particular, the focus is on tabular foundation models which are pretrained on large amounts of tabular data, and are capable of in-context learning (i.e. making predictions based on new labeled data at inference time, without updating the weights). The goal of this paper is to ensure fairness with respect to a sensitive feature corresponding to e.g. a demographic group (using a few classic “fair ML” performance measurements).

The paper presents three relatively simple methods: “correlation remover”, which involves transforming features to reduce correlation between sensitive and non-sensitive features, “uncertain”, which consists in selecting in-context demonstrations which have high uncertainty, and “balanced”, which consists in selecting in-context demonstrations to have a balanced population wrt the sensitive feature.

In their experiments, the authors find that correlation remover performs remarkably poorly (although a variant performs better). The other methods perform reasonably well. The authors also consider the number of demonstration samples, and find that generally, accuracy continues to increase with large demonstration sets, whereas fairness properties actually get slightly worse but plateau quickly. They also compare to regular LLMs (as opposed to tabular foundation models) and find that performance remains competitive, although not always better.

For correlation remover and “uncertain”, there is also a “knob” that can be tuned to trade off fairness and accuracy. There is then a question of what the Pareto frontier between accuracy and fairness looks like. The authors do these experiments, also considering different variations on the "uncertain" method, and different foundation models; unsurprisingly, since “correlation remover” performed so poorly overall, it is Pareto dominated by other methods.

**Additional Comments:**

While TMLR does not consider novelty, this work is very closely adjacent to some other work. For example there are other papers looking at preprocessing for fairness in ICL using some of the same techniques (just with regular LLMs, not with tabular foundation models). This is all cited, and it's not a problem per se, but I would have preferred to see the distinctions and similarities emphasized more clearly.

**Audience:**

Yes

**Audience Explanation:**

Fair ML is of interest to some people in TMLR's audience, as are tabular foundation models, so the combination of the two is also of interest. Anyone who needs to ensure fairness properties are satisfied when using ICL with tabular foundation models will get something out of this paper.

**Broader Impact Concerns:**

No broader impact concerns.

**Claims And Evidence:**

Yes

**Claims Explanation:**

Everything in the paper is described very clearly. The choices of fairness metrics and datasets are totally standard and defensible. The experiments are thorough and convincing, and answer a number of interesting questions about the relative performance of these methods.

**Requested Changes:**

I don't have any substantive requested changes. The paper does what it sets out to do.

Section 2 has one typo that I saw ("Tabular Fondation Models").

---

> ### Author Response · Authors · 2025-10-04
> **Reviewer qoEa**
>
> We thank the reviewer for the positive and encouraging assessment of our work.
> - We corrected the typo in Section 2 and carefully proofread the full paper to fix minor errors.
> - Following the reviewer’s suggestion, we also revised the related work section to more clearly highlight the distinctions and similarities between our study on _tabular foundation models_ and prior work on fairness preprocessing in ICL with _LLMs_. This should make the novelty and contributions of our work more explicit.
>
> We are grateful for the supportive feedback and for pointing out these improvements.

---

### Review · Reviewer_DFr8 · 2025-09-27

**Summary Of Contributions:**

This paper investigates fairness in in-context learning using transformer-based tabular foundation models, evaluating three fairness-enhancing methods—correlation removal, group-balanced sample selection, and uncertainty-based sample selection—on multiple benchmark datasets. The study finds that the uncertainty-based strategy consistently improves group fairness metrics with minimal impact on predictive accuracy.

**Audience:**

Yes

**Audience Explanation:**

Yes, the findings on improving fairness in in-context learning with tabular foundation models would likely interest TMLR’s audience

**Claims And Evidence:**

Yes

**Claims Explanation:**

The submission makes several important claims about improving fairness in in-context learning with tabular foundation models, and the evidence provided is generally accurate and clear.

**Requested Changes:**

1. The correlation remover method unexpectedly amplifies bias. The authors should provide a deeper analysis of why this happens and suggest ways to mitigate it.
2. While the uncertainty-based method shows promise, the paper lacks a detailed explanation of how the uncertainty is calculated and how it specifically reduces bias.
3. The study uses a limited set of datasets. How well do these methods generalize to other types of tabular data, especially those with different distributions or larger scales?
4. The paper does not discuss the computational cost of the proposed methods. How do they scale with the size of the dataset and the complexity of the model?
5. The ablation study on the impact of in-context set size is insightful.

---

> ### Author Response · Authors · 2025-10-04
> **Response to Reviewer DFr8**
>
> We thank the reviewer for the helpful feedback. Below we clarify and expand on the requested changes.
>
> 1. **On correlation remover amplifying bias**
>
> >1. The correlation remover method unexpectedly amplifies bias. The authors should provide a deeper analysis of why this happens and suggest ways to mitigate it.
>
> We provide a deeper analysis in Section 4.5. Bias amplification arises from _sensitive attribute leakage_ when transformations are applied to the test set. As shown in the revised Table 1, sensitive attribute reconstruction reaches nearly 100% accuracy, confirming leakage. To mitigate this, we propose applying correlation removal **only to the training set** (variant S2). Figure 4 and revised Table 2 show that this variant reduces reconstruction accuracy and improves fairness compared to the vanilla baseline.
>
> 2. **On uncertainty-based selection and bias reduction**
>
> > 2. While the uncertainty-based method shows promise, the paper lacks a detailed explanation of how the uncertainty is calculated and how it specifically reduces bias.
>
> Thank you for raising the point. We expanded Section 3 and Appendix B.3 with a concrete example. Briefly, we train a sensitive attribute classifier (LR or TabPFN) on a small fraction of the data, then use conformal prediction to estimate uncertainty. Samples with uncertain predictions (prediction set size equals 2) are kept in the context set, which reduces the information about the sensitive attribute available to the model. As Table 1 shows, this significantly lowers sensitive attribute reconstruction accuracy, explaining the observed fairness improvements.
>
> 3. **Dataset diversity and generalization**
>
> >3. The study uses a limited set of datasets. How well do these methods generalize to other types of tabular data, especially those with different distributions or larger scales?
>
> We evaluated on **8 datasets** across domains (ACS, German Credit, Diabetes, CelebA), with sizes ranging from ~1K (German Credit) to >200K (CelebA). Current foundation models impose some input limits (e.g., TabPFNv2: 10K samples, 500 features), so we subsampled larger datasets where necessary. The consistent trends across datasets suggest good generalization, though larger-scale extensions remain an exciting future direction, even for tabular foundation models.
>
> 4. **Computational cost and scalability**
>
> > 4. The paper does not discuss the computational cost of the proposed methods. How do they scale with the size of the dataset and the complexity of the model?
>
> All fairness interventions are lightweight: group-balanced (sampling), correlation remover (feature-wise regression), uncertainty-based (classifier + conformal prediction). The main cost comes from the foundation model itself. We now report runtime on CelebA and ACSIncome in Appendix C, showing that the Logistic-regression-based uncertainty method is fastest, while TabPFN-based uncertainty incurs overhead. This provides practical guidance for deployment under computational constraints.
>
>
> > 5. The ablation study on the impact of in-context set size is insightful.
>
> We appreciate the positive feedback.
>
>
> We thank the reviewer again for the thoughtful comments, which helped us improve the clarity of the work. We hope the clarification provided addresses the concerns raised.

---

### Review · Reviewer_e8tx · 2025-09-27

**Summary Of Contributions:**

This work conduct a systematic study of fairness mitigation methods for ICL under the context of tabular foundation model. Extensive experiments are performed to study the fairness utility tradeoff between multiple fairness metrics, datasets, and methods. The authors observe that correlation remover could exacerbates the unfairness of ICL.

**Audience:**

Yes

**Audience Explanation:**

The area of ICL using tabular foundation model is interesting and relatively under explored. I think such systematic study is important and is interesting to the community

**Claims And Evidence:**

Yes

**Claims Explanation:**

The claims are well supported. The experiments are extensive.

**Requested Changes:**

- While I like Section 4.2, I think the comparison in Figure 1 is not apple-to-apple comparison and a bit unnecessary. For the following reason: 1. given that we already have the pareto fairness utility tradeoff, it's not clear to me what is the point of selecting a single point from the curve and compare. 2. The authors mention Figure 1 is obtained by fixing $\alpha$ and $\varepsilon$. However, these are hyperparameters for different methods. How are these directly comparable? I recommend removing this section as (correct me if I'm wrong) the information is a direct subset of Section 4.2 and could be confusing.
- The authors report accuracy as utility. However, for tasks such as ACS, one cares about metrics such as precision and recall, could the authors also report these metrics and the pareto frontier between these metrics and fairness?
- The authors should provide more motivation on why group fairness is the main fairness notion rather than other fairness notions such as individual fairness?

---

> ### Author Response · Authors · 2025-10-04
> **Response to Reviewer e8tx**
>
> We thank the reviewer for the constructive feedback. Below we summarize the revisions (highlighted in red) and clarifications made in response to the concerns raised.
>
> 1. **On Figure 1 and Section 4.2**
>
> > While I like Section 4.2, I think the comparison in Figure 1 is not apple-to-apple comparison and a bit unnecessary. For the following reason: [...]  I recommend removing this section as (correct me if I'm wrong) the information is a direct subset of Section 4.2 and could be confusing.
>
> We agree that the fairness–utility tradeoff plots already capture the comparisons of interest. The original Figure 1 was intended as a compact overview, but we recognize it may be redundant and potentially confusing. We have therefore removed Figure 1 and revised Section 4.2 accordingly. All fairness interventions, including the group-balanced baseline, are now consistently compared via Pareto fronts. We believe this revision makes the results section clearer and alleviates any confusion.
>
> 2. **Utility metrics beyond accuracy**
>
> > The authors report accuracy as utility. However, for tasks such as ACS, one cares about metrics such as precision and recall, could the authors also report these metrics and the pareto frontier between these metrics and fairness?
>
> We followed (Ding et al., 2022)[1] in using accuracy for ACS tasks. To broaden evaluation, we have now included **Pareto fronts using ROC AUC** as a utility metric. These results are reported in **Appendix Figures 29 and 30**, showing consistent trends across ACS datasets and baselines.
>
> 3. **Choice of group fairness as the main notion**
>
> > The authors should provide more motivation on why group fairness is the main fairness notion rather than other fairness notions such as individual fairness?
>
> Thank you for raising this point. We clarified in the Introduction that we focus on group fairness because (i) it is the most widely adopted family of metrics in the fairness literature, and (ii) the interventions studied (correlation removal, group-balanced sampling, uncertainty-based sampling) are specifically designed to mitigate group disparities. We also acknowledge in the revised text that **individual and counterfactual fairness** are important and interesting future directions.
>
> We thank the reviewer again for the thoughtful comments, which helped us improve the clarity of the work. We hope the revised paper fully addresses the concerns raised, and we remain available to address any remaining concerns.

---

### Decision · Action_Editor_iceN · 2025-12-12

**Recommendation:** Accept as is

**Audience:**

Yes

**Audience Explanation:**

The topic is timely for TMLR and one interesting line of work would be to investigate whether one could provide theoretical justifications for the observations in the paper.

**Claims And Evidence:**

Yes

**Claims Explanation:**

Fairness is a central challenge for modern ML systems and the paper proposes three pre-processing techniques and one of them (uncertainty-based sample selection) is shown to improve fairness metrics without impacting accuracy.

The reviewers all have recommended acceptance and this is indeed an easy task for the action editor!